# Pharmacologically inducing regenerative cardiac cells by small molecule drugs

Wei Zhou[1,2†], Kezhang He[1†], Chiyin Wang[3], Pengqi Wang[1], Dan Wang[1], Bowen Wang[1,2], Han Geng[1], Hong Lian[3], Tianhua Ma[1*], Yu Nie[3*], Sheng Ding[1,2*]

[1]School of Pharmaceutical Sciences, Tsinghua University, Beijing, China; [2]Tsinghua-Peking Joint Center for Life Sciences, Tsinghua University, Beijing, China; [3]State Key Laboratory of Cardiovascular Disease, Fuwai Hospital, National Center for Cardiovascular Disease, Chinese Academy of Medical Sciences and Peking Union Medical College, Beijing, China

*For correspondence:
matianhua@tsinghua.edu.cn
(TM);
nieyuniverse@126.com (YN);
shengding@tsinghua.edu.cn (SD)

[†]These authors contributed
equally to this work

Reviewing Editor: Nagalingam
R Sundaresan, Indian Institute of
Science, India

## eLife Assessment

This manuscript offers **valuable** information on the combinatory effect of small molecules, CHIR99021 and A-485 (2C), during the reprogramming of mature cardiomyocytes into regenerative cardiac cells on stimulating cardiac cell regeneration. Although the study used several hESC lines and an in vivo model of myocardial injury to demonstrate the regenerative potential of cardiac cells, the manuscript is still **incomplete** as several concerns remain unanswered, including the lack of validation of the conclusions from scRNA-seq. It is still unclear how a small fraction of dedifferentiating cardiac cells can offer such broad effects on regeneration both in vitro and in vivo. If validated, this study might unlock potential therapeutic strategies for cardiac regeneration.

**Abstract** Adult mammals, unlike some lower organisms, lack the ability to regenerate damaged hearts through cardiomyocytes (CMs) dedifferentiation into cells with regenerative capacity. Developing conditions to induce such naturally unavailable cells with potential to proliferate and differentiate into CMs, that is, regenerative cardiac cells (RCCs), in mammals will provide new insights and tools for heart regeneration research. In this study, we demonstrate that a two-compound combination, CHIR99021 and A-485 (2C), effectively induces RCCs from human embryonic stem cell-derived TNNT2[+] CMs in vitro, as evidenced by lineage tracing experiments. Functional analysis shows that these RCCs express a broad spectrum of cardiogenesis genes and have the potential to differentiate into functional CMs, endothelial cells, and smooth muscle cells. Importantly, similar results were observed in neonatal rat CMs both in vitro and in vivo. Remarkably, administering 2C in adult mouse hearts significantly enhances survival and improves heart function post-myocardial infarction. Mechanistically, CHIR99021 is crucial for the transcriptional and epigenetic activation of genes essential for RCC development, while A-485 primarily suppresses H3K27Ac and particularly H3K9Ac in CMs. Their synergistic effect enhances these modifications on RCC genes, facilitating the transition from CMs to RCCs. Therefore, our findings demonstrate the feasibility and reveal the mechanisms of pharmacological induction of RCCs from endogenous CMs, which could offer a promising regenerative strategy to repair injured hearts.

## Introduction

Lower organisms like zebrafish exhibit a remarkable capacity for cardiac regeneration through cardiomyocyte (CM) dedifferentiation, characterized by the reactivation of embryonic cardiogenic genes and disassembly of sarcomeric structures (*Kikuchi et al., 2010*; *Lepilina et al., 2006*). In contrast,

mammals show limited cardiac regeneration, typically restricted to the early postnatal period (PN1 to PN6) in mice, beyond which the regenerative ability sharply declines (*Porrello et al., 2011*). This stark difference underscores the need for novel strategies to induce a regenerative state in adult mammalian CMs, akin to that observed in zebrafish, to overcome the inherent barriers to CM dedifferentiation.

During mammalian heart development, CMs arise predominantly from the first heart field and the second heart field (SHF) (*Srivastava, 2006*), with SHF cells marked by the pioneer transcription factor ISL1 proliferating and differentiating into major cardiovascular cell types, CMs, smooth muscle cells (SMCs), and endothelial cells (ECs) during embryonic development (*Bu et al., 2009*; *Moretti et al., 2006*). Notably, ISL1[+] cells in neonatal mouse hearts show potential for expansion and differentiation into CMs under conducive in vitro conditions (*Laugwitz et al., 2005*). Given that ISL1 transcriptionally governs the expression of a suite of cardiac genes, such as *NKX2-5* (*Ma et al., 2008*), *FGF10* (*Watanabe et al., 2012*), which are essential for embryonic cardiogenesis, inducing ISL1 expression in CMs could be a critical determinant in the induction of cells possessing regenerative capacity.

Aiming to replicate this regenerative capacity, our study focused on identifying small molecules capable of inducing ISL1 expression in CMs. Through a combinatorial screening approach, we discovered a novel combination of CHIR99021 and A-485, or 2C, that efficiently and unprecedentedly induces dedifferentiation of human CMs into regenerative cardiac cells (RCCs). These RCCs exhibited disassembled sarcomeric structures, high expression of embryonic cardiogenic genes, and an increased number of CMs through re-differentiation in vitro. Further investigations showed that 2C robustly generates RCCs in adult mouse hearts and improves cardiac function in mice experiencing myocardial infarction (MI). This proof-of-concept discovery demonstrates that a simple combination of small molecule drugs can endow the mammalian heart with regenerative capacity by reprogramming CMs into RCCs.

## Results

### Efficient induction of dedifferentiation in human embryonic stem cell-derived CMs by 2C treatment

To obtain an adequate number of CMs for small molecule screening, we differentiated human embryonic stem cells (hESCs) into CMs (*Video 1*, *Figure 1—figure supplement 1A*) following a well-established stepwise protocol (*Lian et al., 2013*). Nearly homogeneous contracting CMs were observed on day 10 of differentiation, consistent with previous reports. After purification with glucose-depleted medium containing abundant lactate (*Figure 1—figure supplement 1B*), highly pure TNNT2[+] CMs were obtained. These were subsequently dissociated and seeded into 96-well plates. Once the CMs resumed contraction, they were treated with individual small molecules from a collection of over 4000 compounds for 3 days (*Figure 1—figure supplement 1C*), and then fixed and immunostained for ISL1. Using a high-content imaging and analysis system, five compounds were initially identified as potential inducers of dedifferentiation in CMs, indicated by induced sarcomere disassembly and ISL1 expression (*Figure 1—figure supplement 1D*). Further comparison revealed that the unique combination of CHIR99021 and A-485 (2C) most efficiently induced ISL1 expression in TNNT2[+] CMs (*Figure 1—figure supplement 1E*). Notably, while A-485 was screened at a concentration of 10 µM, its optimal working concentration in subsequent experiments was determined to be 0.5 µM based on titration experiments (*Figure 1—figure supplement 1F, G*). Of note, compound I-BET-762 also showed some capacity to induce ISL1 expression. However, it was less effective than A-485 in combination with CHIR99021 (*Figure 1—figure supplement 1H*).

When CMs with well-organized sarcomeres were treated with 2C in vitro, cells started to exhibit a dedifferentiation-associated phenotype,

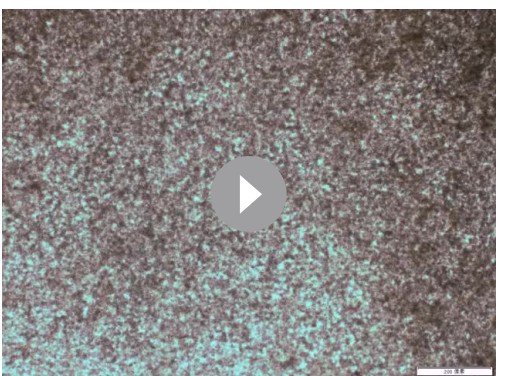

**Video 1.** Contracted cardiomyocytes (CMs) at SD4. Human embryonic stem cell (hESC)-derived CMs at day 4 (SD4) after lactate selection.
https://elifesciences.org/articles/93405/figures#video1

such as reduction of cell size after 24 hr, growing as clusters after 48 hr, and forming colonies after 60 hr (*Figure 1A* and *Figure 1—figure supplement 2A*). During this period, the expression of TNNT2 and MYL2 gradually downregulated while ISL1 expression and the percentage of ISL1+ cells increased (*Figure 1—figure supplement 2B*). Compared to untreated CMs, significant sarcomere disassembly and reduction of cytoplasmic/nuclear area were observed in cells treated with 2C for 60 hr (*Figure 1B, C*). In parallel, the number and percentage of TNNT2+ cells remained constant while ISL1+ cells increased remarkably (*Figure 1D, E* and *Figure 1—figure supplement 2C*), accompanied by a ~threefold increase in ISL1 expression at both mRNA and protein levels (*Figure 1F* and *Figure 1—figure supplement 2B*). Other early embryonic cardiogenesis genes, including *MESP1*, *LEF1* (*Klaus et al., 2007*), *FUT4* (*Wang et al., 2019*), and *NR2F2* (*Churko et al., 2018*), were also highly expressed following 2C treatment (*Figure 1G* and *Figure 1—figure supplement 2D*). Furthermore, 2C treatment significantly decreased the expression of CM-specific genes, such as *TNNT2*, *MYL2*, *MYL7*, and *MEF2C*, while pan-cardiac transcription factors *GATA4*, *TBX5*, and *NKX2*-5 showed less effect (*Figure 1G* and *Figure 1—figure supplement 2D*). Notably, 2C-induced ISL1 expression even in mature CMs derived from hESCs treated with ZLN005 (*Liu et al., 2020*; *Figure 1—figure supplement 2E*). These results indicate that 2C treatment effectively reprograms CMs toward a dedifferentiated, embryonic-like state.

## 2C-induced cells possess regenerative capacity

To confirm the regenerative capacity of 2C-induced cardiac cells, we investigated cell proliferation using a BrdU incorporation assay. Immunostaining showed theco-localization of ISL1 and BrdU labeling in these cells (*Figure 2A*). Statistical analysis revealed a significant decrease in the nuclear area and an increase in the number of ISL1+/BrdU+-positive cells following 2C treatment (*Figure 2B, C*). We then examined the potential for re-differentiation of 2C-induced cardiac cells into CMs. As expected, upon withdrawal of 2C and subsequent culture in CM media for 3 days (60h+3d), we observed the emergence of spontaneously contracting cells exhibiting CM-specific morphology and typical cytoplasmic/nuclear area size (*Video 2* and *Figure 2D, E*). These cells downregulated early embryonic cardiogenesis genes and upregulated CM-specific genes, with no noticeable changes in pan-cardiac genes (*Figure 2—figure supplement 1A*). Remarkably, there was a notable increase (roughly 1.4-fold) in the number of re-differentiated CMs with clear TNNT2 staining relative to the initial CMs before 2C treatment (*Figure 2F*), demonstrating the regenerative potential in 2C-induced cardiac cells. Additionally, 2C-induced cardiac cells could differentiate into SMA+ SMCs or CD31+/VE-Cadherin+ ECs in the presence of PDGF-BB and TGF-β1 or VEGF, bFGF, and BMP4, respectively (*Figure 2G, H* and *Figure 2—figure supplement 1B*). In contrast, these genes expressed were not detectable in DMSO-treated cells. Collectively, our findings indicate that 2C treatment facilitates the conversion of CMs into cardiac cells with regenerative capability, including proliferative potential and differentiation into the three major cardiovascular cell types, thus named RCCs.

## Lineage tracing demonstrated that 2C-induced RCCs dedifferentiated from TNNT2+ CMs

Although TNNT2+ CMs purified using a lactate-based culture medium were nearly homogeneous populations, a small fraction (<4%) of purified cells still expressing ISL1 (*Figure 1—figure supplement 1B*). To ascertain that the RCCs induced by 2C treatment were indeed dedifferentiated from TNNT2+ CMs rather than originating from the proliferation of residual ISL1+ cells, we conducted lineage-tracing experiments. Using an ISL1mCherry/+ H9 hESC (K9) line established through CRISPR-based knock-in (*Figure 3—figure supplement 1A–D*), we verified that mCherry expression accurately mirrored endogenous ISL1 expression (*Figure 3A–C*). Following the purification of K9-derived CMs and the selection of mCherry-negative cells by fluorescence-activated cell sorting (FACS; *Figure 3D, E*), these cells were confirmed to be TNNT2+ CMs (*Figure 3—figure supplement 1E*). Upon treatment of these mCherry-negative CMs with 2C for 60 hr, we observed remarkable morphological changes in mCherry+/ISL1+ RCCs compared to mCherry−/TNNT2+ CMs (*Figure 3F*), accompanied by a dramatic downregulation of TNNT2 and MYL2 expression and a notable upregulation of ISL1 and mCherry expression (*Figure 3G*). FACS analysis further validated that 2C could convert K9-derived mCherry-negative CMs into mCherry+/ISL1+ cells (*Figure 3H, I*). Similar findings were obtained using

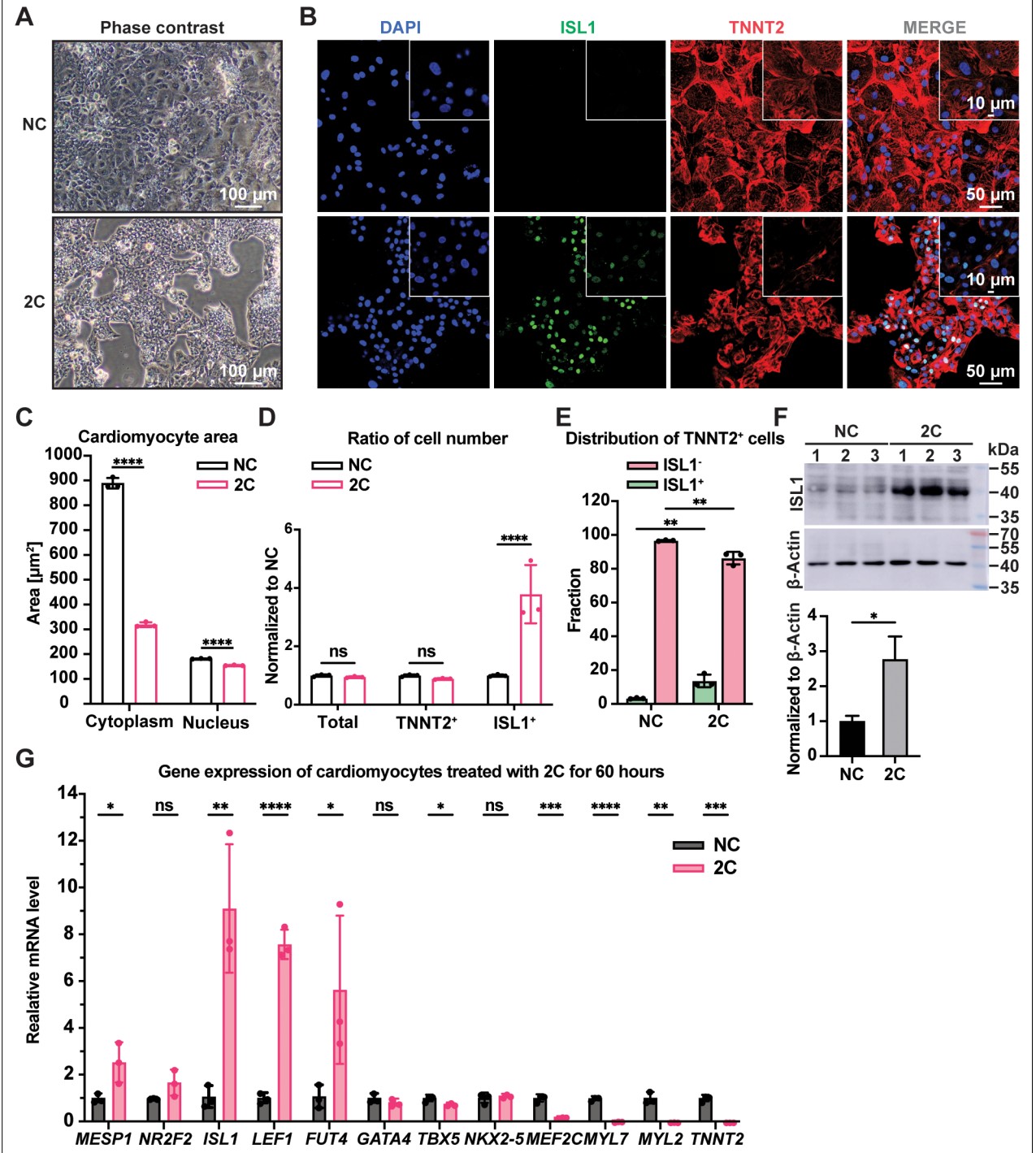

**Figure 1.** 2C treatment-induced dedifferentiation of human embryonic stem cell (hESC)-derived cardiomyocytes (CMs) toward ISL1-expressing cells. Cells induced from hESC-derived CMs by treatment with dimethyl sulfoxide (DMSO) for 60 hr. (**A**) Phase contrast images showing cell morphology. (**B**) Immunofluorescence staining of the ISL1 (ISL1, green), and the CM marker cardiac troponin T (TNNT2, red) in the cells. Nuclei were stained by DAPI (4',6-diamidino-2-phenylindole) and presented in DNA blue. (**C**) Cytosolic and nuclear areas of the cells. Data are shown as mean ± SD ($n$ = 3 independent experiments, represented as dots). Two-way ANOVA with Dunnett's multiple comparisons test. ****$p < 0.0001$. (**D**) Changes in cell number after 2C treatment. Total cell number, TNNT2+ cell number, and ISL1+ cell number were normalized to the negative control DMSO (NC). Data are shown as mean ± SD ($n$ = 3 independent experiments, represented as dots). Two-way ANOVA with Šidák's multiple comparisons test. ns, not significant ($p >$ 0.05), ****$p < 0.0001$. (**E**) Fraction of ISL1+ cells in TNNT2+ cells. Data are shown as mean ± SD ($n$ = 3 independent experiments, represented as dots). Two-way ANOVA with Šidák's multiple comparisons test. **$p < 0.01$. (**F**) Western blot and quantitative analysis of ISL1 expression in DMSO (NC) or 2C-treated CMs for 60 hr. Data are shown as mean ± SD. Unpaired $t$ test. *$p < 0.05$. (**G**) Relative gene expression of embryonic cardiogenesis marker genes

*Figure 1 continued on next page*

*Figure 1 continued*

(*MESP1*, *ISL1*, *NR2F2*, *FUT4*, and *LEF1*), pan-cardiac genes (*GATA4*, *TBX5*, and *NXK2-5*), and CM marker genes (*MEF2C*, *TNNT2*, *MYL2*, and *MYL7*) in the cells treated by DMSO (NC) or 2C for 60 hours (60 h). Data are shown as mean ± SD (*n* = 3 independent experiments, represented as dots). Multiple unpaired *t* tests. ns, not significant (p > 0.05), *p < 0.05, **p < 0.01, ***p < 0.001, ****p < 0.0001.

The online version of this article includes the following source data and figure supplement(s) for figure 1:

**Source data 1.** PDF file containing original western blots for *Figure 1F*, indicating the relevant bands and treatments.

**Source data 2.** Original files for western blot analysis displayed in *Figure 1F*.

**Figure supplement 1.** 2C-induced reprogramming of ISL1⁺ cells from human embryonic stem cell (hESC)-derived cardiomyocytes (CMs).

**Figure supplement 2.** 2C treatment gradually induced dedifferentiation of cardiomyocytes (CMs).

the HUES7 hESC line with an ISL1-mCherry knock-in reporter (K7) (*Figure 3—figure supplement 2*), robustly demonstrating that 2C induces ISL1 expression in TNNT2⁺ CMs, effectively generating RCCs.

Additionally, we used a lineage-tracing system to unequivocally confirm the conversion of CMs to ISL1-expressing RCCs by 2C treatment. In this assay, CMs were transfected with plasmids encoding CreERT2 under the control of the TNNT2 promoter and an EGFP reporter following a flox-stop-flox cassette (*Figure 3—figure supplement 3A*). After optimizing viral titers and infection timing (*Figure 3—figure supplement 3B–D*), approximately 0.6% of K9-derived mCherry-negative CMs were permanently labeled with EGFP following 6 days of tamoxifen treatment (*Figure 3J, K*). These EGFP-labeled mCherry-negative CMs were then sorted by flow cytometry and treated with 2C for 60 hr. As expected, ISL1 expression was observed in these EGFP⁺ CMs (*Figure 3L*). Collectively, these results strongly indicate that the RCCs with ISL1 expression were genuinely generated by 2C treatment from TNNT2⁺ CMs.

## 2C-induced dedifferentiation of CMs provided a protective effect on cardiac infarction

To examine the reprogramming effect of 2C on CMs in vivo, we first tested 2C on primary neonatal rat CMs (*Sakurai et al., 2014*), which consistently induced RCC generation with corresponding Isl1 induction and morphological changes (*Figure 4—figure supplement 1A*). We then evaluated whether 2C could reprogram endogenous CMs to RCCs in vivo. Neonatal SD rats were intraperitoneally administered 20 mg/kg of CHIR99021 and 10 mg/kg of A-485 daily for 5 days (*Figure 4—figure supplement 1B*). Compared to the vehicle/DMSO controls (NC), Isl1 was robustly induced in the CMs of rats treated with 2C without apparent effects on heart to body weight ratio (*Figure 4—figure supplement 1C, D*). These induced Isl1-expressing cells were widely distributed within the region 600–3000 µm down from the base of the hearts, including the aorta (Ao), left/right atria, and both ventricles (*Figure 4—figure supplement 1E*). All these Isl1⁺ cells also expressed Tnnt2, indicating that 2C-induced RCCs originated from CMs in vivo. Importantly, neither CHIR99021 nor A-485 alone induced Isl1⁺ cells in ventricles, highlighting the combined effect of 2C on RCC induction (*Figure 4—figure supplement 2*).

Similarly, RCCs were efficiently induced in the Ao root and RA regions of the heart dissected from adult 129SvJ mice administered 2C (20 mg/kg of CHIR99021 and 10 mg/kg of A-485) for 5 consecutive days (*Figure 4A*). To explore whether 2C-induced RCCs could rescue cardiac function in mice subjected to MI, we conducted both prophylactic and therapeutic studies. In the prophylactic setting, mice were pre-treated with either 2C or vehicle/DMSO for 5 days (*Figure 4B*) before inducing MI by ligation of left anterior descending (LAD) artery. When measuring cardiac function with magnetic resonance imaging on days 1, 8, 25, and 35 post-MI, we found pretreatment with 2C significantly improved both cardiac function and survival rate without affecting body weight (*Figure 4C, E*). In the therapeutic setting (*Figure 4F*), mice treated with 2C post-MI displayed recovered cardiac function as assessed by left ventricular ejection fraction and echocardiography (*Figure 4G, H*). Cardiac fibrosis was largely ameliorated in 2C-treated mice, displaying significantly smaller scar sizes compared to control mice (*Figure 4I*). Altogether, these in vivo studies collectively indicate that 2C-induced RCCs possess regenerative capacities comparable to those generated in vitro, shedding light on the development of small molecule drugs with similar mechanisms for therapeutic use in repairing or regenerating hearts after cardiac injury.

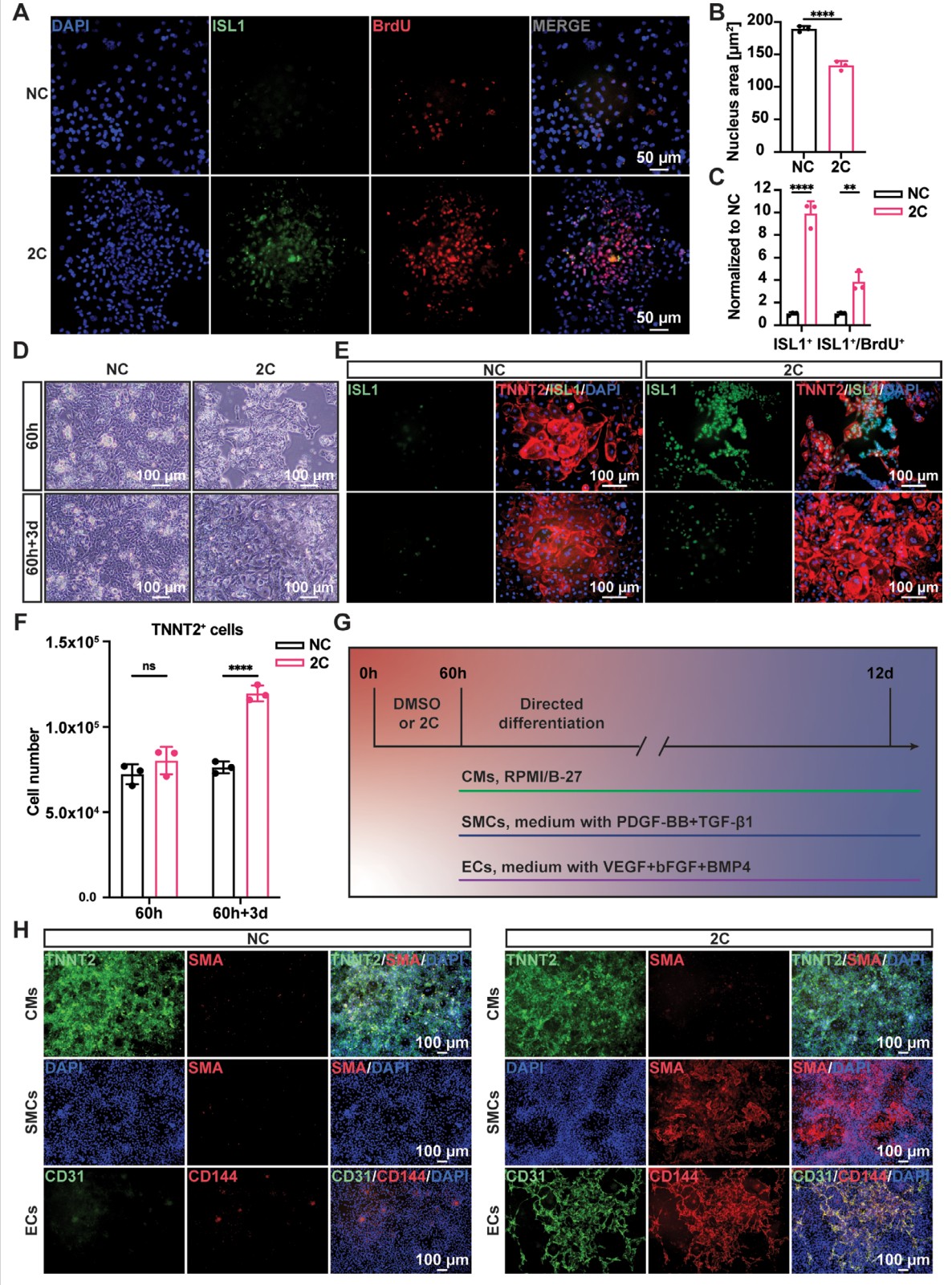

**Figure 2.** Regenerative ability of 2C-induced regenerative cardiac cells (RCCs). Immunofluorescence staining (**A**) and statistical analysis (**B, C**) of the ISL1 (green) and BrdU (red) double positive RCCs induced from cardiomyocytes (CMs) by treatment with DMSO (NC) or 2C for 60 hr. The ISL1+ cell number and ISL1+/BrdU+ cell number were normalized to the negative control DMSO (NC). DAPI (4′,6-diamidino-2-phenylindole) staining labeled nuclei as blue. Data are shown as mean ± SD ($n$ = 3 independent experiments, represented as dots). Multiple unpaired $t$ tests in (**B**), ****p < 0.0001. Two-way ANOVA

*Figure 2 continued on next page*

*Figure 2 continued*

with Šidák's multiple comparisons test in (**C**), **p < 0.01, ****p < 0.0001. (**D**) Phase contrast images of human embryonic stem cell (hESC)-derived CMs treated by DMSO (NC) or 2C for 60 hour (60 h) and subsequently cultured in the absence of 2C for another 3 days (60h+3d). (**E**) Immunostaining showed the expression of ISL1 (green) and TNNT2 (red) in the cells under the same condition in (**D**). (**F**) Statistical analysis of TNNT2+ cell numbers under the same condition in (**D**). Data are shown as mean ± SD (n = 3 independent experiments, represented as dots). Two-way ANOVA with Šidák's multiple comparisons test. ns, not significant (p > 0.05), ****p < 0.0001. (**G**) Schematic diagram of directed differentiation of 2C-induced RCCs toward cardiomyocytes (CMs), smooth muscle cells (SMCs), and endothelial cells (ECs). (**H**) Immunostaining showed the expression of EC markers (CD31, green and CD144, red), SMC marker (SMA, red), and CM marker (TNNT2, green). DAPI (4′,6-diamidino-2-phenylindole) staining labeled nuclei as blue.

The online version of this article includes the following figure supplement(s) for figure 2:

**Figure supplement 1.** 2C-induced regenerative cardiac cells (RCCs) represent the differentiation potential ability.

## Dedifferentiation requirement of CHIR99021 and A-485

To elucidate the mechanisms behind the in 2C-induced dedifferentiation of CMs into RCCs, we performed bulk RNA-seq on K9-derived ISL1/mCherry-negative CMs treated with either DMSO as a negative control (NC) or the 2C combination for 60 hr (*Figure 5A*). Analysis of differentially expressed genes (DEGs) revealed substantial transcriptomic alterations; specifically, 2C treatment upregulated embryonic cardiogenesis genes such as *MSX1*, *BMP4*, *TCF4*, and *LEF1*, and downregulated genes associated with CMs (*Figure 5B, C*). Gene ontology (GO) analysis further indicated a suppression of genes involved in cardiac maturation and muscle contraction, contrasting with an enrichment in genes part of the catenin complex (*Figure 5D–F*). These findings were corroborated by qPCR, confirming the significant upregulation of embryonic genes (e.g., *MSX2*, *NKD1*, *PDGFC*, and *CTNNA2*) in K9-derived mCherry-negative CMs (*Figure 5G*).

We investigated the individual and combined effects of CHIR99021 and A-485 on inducing RCC characteristics. Morphological assessments showed that CHIR99021 significantly reduced both cytoplasmic and nuclear areas—a feature characteristic of RCCs—more effectively than A-485, which primarily reduced cytoplasmic area only (*Figure 5—figure supplement 1A–C*). Sarcomere disassembly was more pronounced with CHIR99021 treatment, aligning with its role in enhancing embryonic cardiogenesis genes like BMP4 and LEF1 (*Figure 5—figure supplement 1D, E*). A-485, while not affecting the nuclear area to the same extent, played a crucial role in the epigenetic reprogramming of CMs. It facilitated the suppression of CM-specific genes, such as *TNNT2*, through epigenetic modifications, enhancing the expression of RCC-specific genes such as *ISL1* in conjunction with CHIR99021 (*Figure 5—figure supplement 1D*). This indicates that while CHIR99021 drives the initial dedifferentiation process by altering the cellular morphology and gene expression associated with embryonic cardiogenesis, A-485 enhances these effects by providing necessary epigenetic conditions for sustaining and stabilizing the RCC phenotype. Despite both compounds independently inducing reduced expression of *TNNT2* and *MYL2*, as well as increased ISL1 expression, CHIR99021 uniquely influences a cohort involved in embryonic cardiogenesis, including *BMP4*, *NKD1*, *MSX2*, *PDGFC*, *LEF1*, and *TCF4* (*Figure 5—figure supplement 1D*). Upon withdrawal of CHIR99021, A-485 or 2C, the cell percentage of ISL1+ cell was similarly and significantly reduced (*Figure 5—figure supplement 1F*). Notably, a drastic increase in the number of TNNT2+ CMs was observed only during the re-differentiation of 2C-treated cells, as confirmed by statistical analysis (*Figure 5—figure supplement 1G*). These findings suggest that while CHIR99021 plays a leading role in 2C-induced dedifferentiation of CMs to RCCs, A-485 is also indispensable, particularly in modifying the epigenetic landscape to support the transition and maintenance of reprogrammed state.

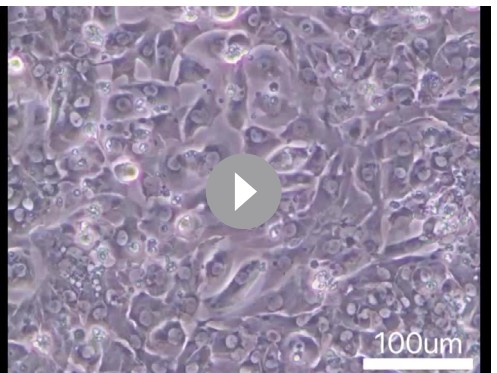

**Video 2.** Contracted cardiomyocytes (CMs) at 60h+3d-2C. Human embryonic stem cell (hESC)-derived CMs treated by 2C for 60 hours (60 h) and subsequently cultured in the absence of 2C for another 3 days (60h+3d).

https://elifesciences.org/articles/93405/figures#video2

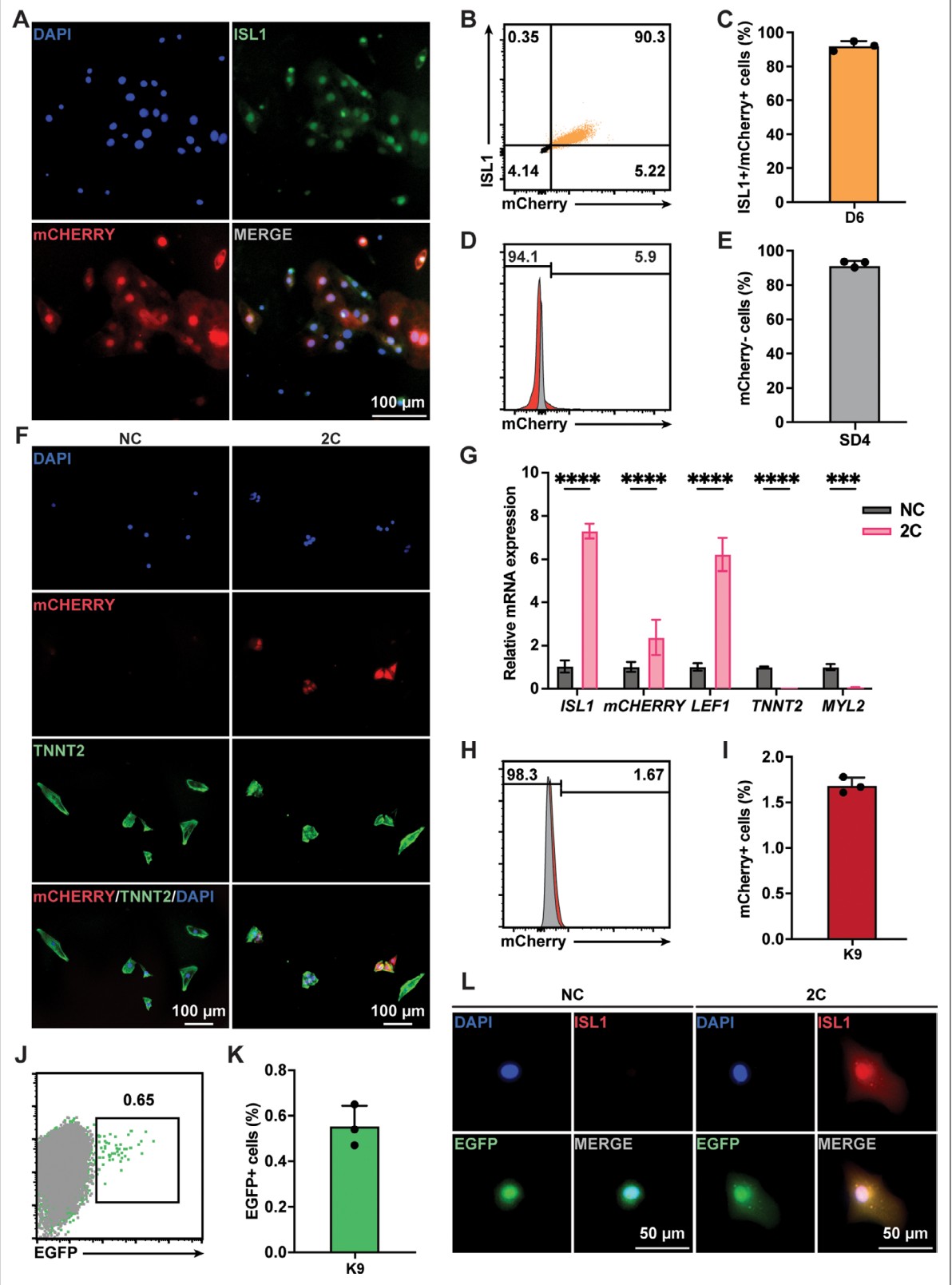

**Figure 3.** Lineage tracing demonstrated 2C-induced dedifferentiation of TNNT2+ cardiomyocytes (CMs) to ISL1-expressing regenerative cardiac cells (RCCs). (**A**) Immunofluorescence images showing expression of endogenous ISL1 (green) and ISL1-mCherry (red) reporter in the cells differentiated from K9 human embryonic stem cell (hESC) KI reporter line at day 6 (D6). DAPI (4′,6-diamidino-2-phenylindole) staining labeled nuclei as blue. (**B, C**) Flow cytometry analysis of the percentage of mCherry+/ISL1+ cells in the cells differentiated from K9 at D6. (**D, E**) Flow cytometry analysis of the percentage

*Figure 3 continued on next page*

*Figure 3 continued*

of mCherry-negative cells at selection day 4 (SD4) in lactate purification medium. (**F**) Cells induced from mCherry-negative CMs by treatment with or without 2C for 60 hr. Images showing the expression of mCHERRY (red) and TNNT2 (green) in the cells. (**G**) Relative gene expression of *ISL1*, *mCHERRY*, *LEF1*, *TNNT2*, and *MYL2* in K9-derived mCherry-negative CMs treated with DMSO (NC) or 2C for 60 hr. Data are shown as mean ± SD (*n* = 2 independent experiments with 4 replicates each). Two-way ANOVA with Šidák's multiple comparisons test. ***p < 0.001, ****p < 0.0001. (**H, I**) Flow cytometry analysis of the percentage of mCherry-positive cells induced from mCherry-negative CMs by treatment with or without 2C for 60 hr. Data are shown as mean ± SD (*n* = 3 independent experiments, represented as dots). Flow cytometric plots showing EGFP-labeled CMs by lineage-tracing of K9-derived mCherry-negative CMs (**J**), and bar graph showing the percentage of mCherry-negative CMs expressing EGFP (**K**). Data are shown as mean ± SD (*n* = 3 independent experiments). (**L**) Images showing the expression of ISL1 (red) and EGFP (green) in the cells induced from EGFP-positive/ mCherry-negative CMs by treatment with or without 2C for 60 hr. DAPI (4',6-diamidino-2-phenylindole) staining labeled nuclei as blue.

The online version of this article includes the following source data and figure supplement(s) for figure 3:

**Figure supplement 1.** Construction of ISL1mCherry/+knock in H9 human ESC line by CRISPR–Cas9.

**Figure supplement 1—source data 1.** PDF file containing original gel image for *Figure 3—figure supplement 1B*, indicating the relevant bands.

**Figure supplement 1—source data 2.** Original files for gel image displayed in *Figure 3—figure supplement 1B*.

**Figure supplement 2.** 2C-induced K7-derived mCherry-negative cardiomyocytes (CMs) into ISL1/mCherry-positive cells.

**Figure supplement 3.** 2C-induced ISL1 expression in cardiomyocytes (CMs) with EGFP labeled by lineage tracing.

## Reprogramming of CMs to RCCs by 2C went through an intermediate cell state

In the exploration of CM reprogramming to RCC using 2C treatment, single-cell RNA sequencing (scRNA-seq) was utilized to trace the progression through distinct cellular states of K9-derived mCherry-negative CMs treated with either DMSO (NC) or the 2C for 60 hr. Through Uniform Manifold Approximation and Projection (UMAP) analysis, we identified seven distinct clusters, observing notable increases in the proportions of cells in clusters 0, 2, and 3 following 2C treatment (*Figure 6A, B*). Cells within cluster 2 prominently expressed genes characteristic of RCCs, such as *ISL1*, *BMP4*, and *FGF20*—a gene crucial for the expansion of early embryonic progenitor cells (*Cohen et al., 2007*), along with the cell proliferation marker *MKI67* (*Figure 6C, D*). This distinct expression pattern marked them as transitioning toward an RCC phenotype. In contrast, clusters 0, 3, 4, and 5 maintained high expression levels of CM-specific markers like *MYH6* and *MYL2* (*Figure 6C, D*), indicating their retention of a mature CM identity. Interestingly, cells in clusters 1 and 6 exhibited characteristics of intermediate cells (ICs), expressing both dedifferentiation markers such as *ACTA2* and genes linked to CM development, including *COL1A1*, *COL1A2*, and *COL3A1* (*Cui et al., 2020*; *Mononen et al., 2020*; *Figure 6C, D*). This highlighted these clusters as transitional states in the reprogramming process from CMs to RCCs.

Further detailed analysis of cluster 2, comparing 435 cells from DMSO-treated samples with 410 cells from 2C-treated samples, allowed us to identify a prominent subcluster (subcluster 0) comprising 408 cells (48.3% of the cluster), exclusively found in 2C-treated samples (*Figure 6E, F*). Cells in this subcluster were defined by high expression of key embryonic cardiogenesis genes such as *MSX1*, *LEF1*, *BMP4*, *MSX2*, and *HAND1*, along with genes typically co-expressed with *ISL1* in SHF progenitors, including *NR2F2*, *TBX5*, *ALCAM* (*Ghazizadeh et al., 2018*), and *CXCR4* (*Andersen et al., 2018*; *Figure 6G*). This specific gene expression profile robustly indicates that the cells in subcluster 0 have adopted an RCC identity, indicative of successful induction by 2C treatment. Thus, a unique gene set that included LIX1, NKD1, PDGFC, ARL4A, AXIN2, FZD7, ISL1, MSX1, MSX2, BMP4, LEF1, and HAND1 was found to determine RCC state during the reprogramming of CMs by 2C, and hence designated as RCC genes.

To further quantify this transition, pseudo-time analysis was performed, clearly delineating the 2C-induced reprogramming trajectory from CMs, highly expressing *TNNT2*, through ICs, characterized by the expression of *COL1A1*, to RCCs, marked by elevated levels of *ISL1* and *FGF20* (*Figure 6H–J*). This analysis confirms the dynamic and stepwise nature of cellular transformation under 2C treatment.

## Epigenetic regulation of CMs and RCCs genes by 2C treatment

As previously documented (*Lasko et al., 2017*), A-485 functions as a p300 acetyltransferase inhibitor, specifically reducing acetylation at lysine 27 of histone H3 (H3K27Ac), but not at lysine 9 (H3K9Ac), as shown in studies with PC3 cells. Our data corroborate these findings, showing a notable decrease in

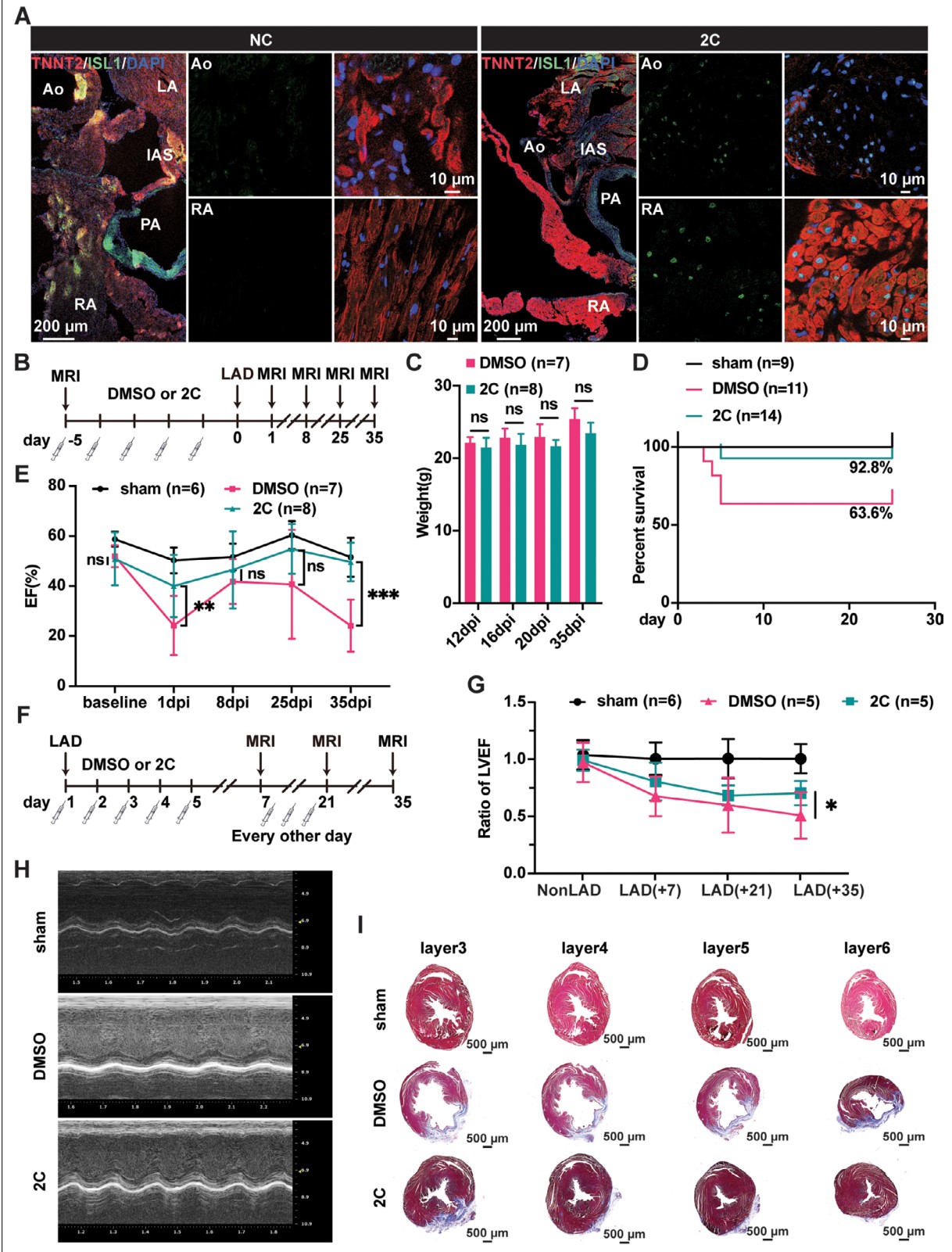

**Figure 4.** Heart regeneration via 2C-induced dedifferentiation of cardiomyocytes (CMs). (**A**) Immunofluorescence staining of ISL1 (green) and TNNT2 (red) in cross-sectioned hearts from 2C or vehicle (DMSO)-treated (NC) adult 129SvJ mice. Ao, aorta. PA, pulmonary artery. LA, left atrial. RA, right atrial. IAS, interatrial septum. DAPI (4',6-diamidino-2-phenylindole) staining labeled nuclei as blue. (**B**) Schematic illustration of the method used to examine the prophylactic effect of 2C in 129SvJ mice post myocardial infarction (MI). (**C**) Body weight of mice pre-treated with vehicle (DMSO) or 2C as shown in

*Figure 4 continued on next page*

*Figure 4 continued*

(**B**) at day 12, day 16, day 20, and day 35 after MI (dpi). Error bars represent SD. ns, not significant (p > 0.05). (**D**) Survival curve of sham-operated mice and mice pre-treated with vehicle (DMSO) or 2C as shown in (**B**), at indicated time points before or after MI. (**E**) Ejection fraction (EF) of sham-operated mice and mice pre-treated with vehicle (DMSO) or 2C as shown in (**B**), before MI (baseline) or at day 1, day 8, day 25, and day 35 after MI (dpi). Data are shown as mean ± SD. Two-way ANOVA with Tukey's multiple comparisons test. ns, not significant (p > 0.05), **p < 0.01, ***p < 0.001. (**F**) Schematic illustration of the method used to examine therapeutic effect of 2C in the 129SvJ mice post MI. (**G**) Serial fMRI measurements showing the cardiac function from sham-operated mice and mice treated with vehicle (DMSO) or 2C at as shown in (**F**). Data are shown as mean ± SD. Two-way ANOVA with Tukey's multiple comparisons test. *p < 0.05. (**H**) Echocardiography of sham-operated mice and mice treated with vehicle (DMSO) or 2C as shown in (**F**) at day 35 post MI. (**I**) Masson staining of serial transverse sections of hearts from sham-operated mice and mice treated with vehicle (DMSO) or 2C as shown in (**F**) at day 35 post MI.

The online version of this article includes the following figure supplement(s) for figure 4:

**Figure supplement 1.** 2C-induced expression of ISL1 in neonatal rats cardiomyocytes (CMs) in vitro and in vivo.

**Figure supplement 2.** Administration of CHIR99021 or A-485 alone cannot induce ISL1 expression in neonatal rat cardiomyocytes (CMs) in vivo.

H3K27Ac levels in CMs treated with A-485 alone, while levels of H3K9Ac remain unaffected (*Figure 7—figure supplement 1A–E*). However, treatment with CHIR99021 or the combined 2C regime did not significantly alter the acetylation patterns of these histone marks (*Figure 7—figure supplement 1*). Intriguingly, we observed that although the number of H3K9Ac$^+$ cells remained unchanged in CMs, the fluorescence intensity was significantly decreased after A-485 treatment (*Figure 7—figure supplement 1F*), suggesting a distinctive regulatory role of A-485 in modulating histone acetylation in CM reprogramming.

To explore the molecular mechanisms by which the 2C treatment modulates gene expression during the reprogramming of hESC-derived CMs into RCCs, we conducted chromatin immunoprecipitation followed by sequencing (ChIP-seq) analyses. This study focused on H3K27Ac and H3K9Ac modifications across various treatment conditions (*Figure 7A*). Our analysis identified regions near transcription start sites (±4 kb) that were rich in H3K4me3, indicative of active promoters in both control (DMSO-treated) and 2C-treated cells (*Figure 7B, C*). Following 2C treatment, there was a marked increase in H3K9Ac and H3K27Ac levels around the transcription start sites, suggesting enhanced gene activity. Specifically, 2C treatment significantly increased H3K9Ac enrichment at 4485 gene promoters and H3K27Ac at 2560 gene promoters, with corresponding decreases in 346 and 1834 genes, respectively (*Figure 7—figure supplement 2A–F*). Notably, genes that showed reduced acetylation in response to A-485 also displayed similar trends under 2C treatment, highlighting genes with functions crucial to cardiac muscle activity such as contraction and sarcomere organization (*Figure 7—figure supplement 2*). In comparison with treatments using A-485 or CHIR99021 alone, the 2C combination notably augmented acetylation at genes involved in cell cycle regulation and cell division, suggesting a synergistic effect of the dual treatment in promoting gene activation essential for cardiac regeneration (*Figure 7—figure supplement 2*).

We also specifically analyzed the acetylation peaks at promoters of key CM genes and RCC genes. A-485 treatment predominantly decreased H3K9Ac and H3K27Ac peaks at CM gene promoters such as *TNNT2*, *TNNI1*, *MYL7*, *MYH6*, and *MYH7*, aiding their downregulation. These reductions were intensified under 2C treatment, reflecting its potent effect in repressing mature CM markers while activating regenerative pathways (*Figure 7D* and *Figure 7—figure supplement 3*). In contrast, CHIR99021 effectively upregulated H3K9Ac and H3K27Ac at promoters of the RCC genes such as *LEF1*, *AXIN2*, *BMP4*, *LIX1*, *MSX1*, *MSX2*, and *NKD1*, with 2C treatment further enhancing these effects, confirming its robust impact on inducing a regenerative phenotype in treated cells (*Figure 7D* and *Figure 7—figure supplement 3*).

Moreover, the distribution of H3K9Ac and H3K27Ac changes during 2C-induced dedifferentiation revealed that these modifications are more dynamically associated with H3K9Ac, particularly in the transition of CMs to RCCs. This dynamic regulation was illustrated by the significant number of genes showing exclusive enrichment under 2C treatment compared to control or single-drug treatments (*Figure 7E, F*). In detail, of genes with H3K9Ac enrichment, 2716 (or 46.1%) out of 5891 annotated genes were only enriched in the cells treated with 2C, while only 527 (or 4.9%) out of 10,785 annotated genes with H3K27Ac enrichment were exclusively observed (*Figure 7E, F*). To further confirm this notion, we performed a comparative analysis of annotated genes between the cells treated with 2C or DMSO to validate the most significant differential genes (i.e., Log10LR >3) under the respective

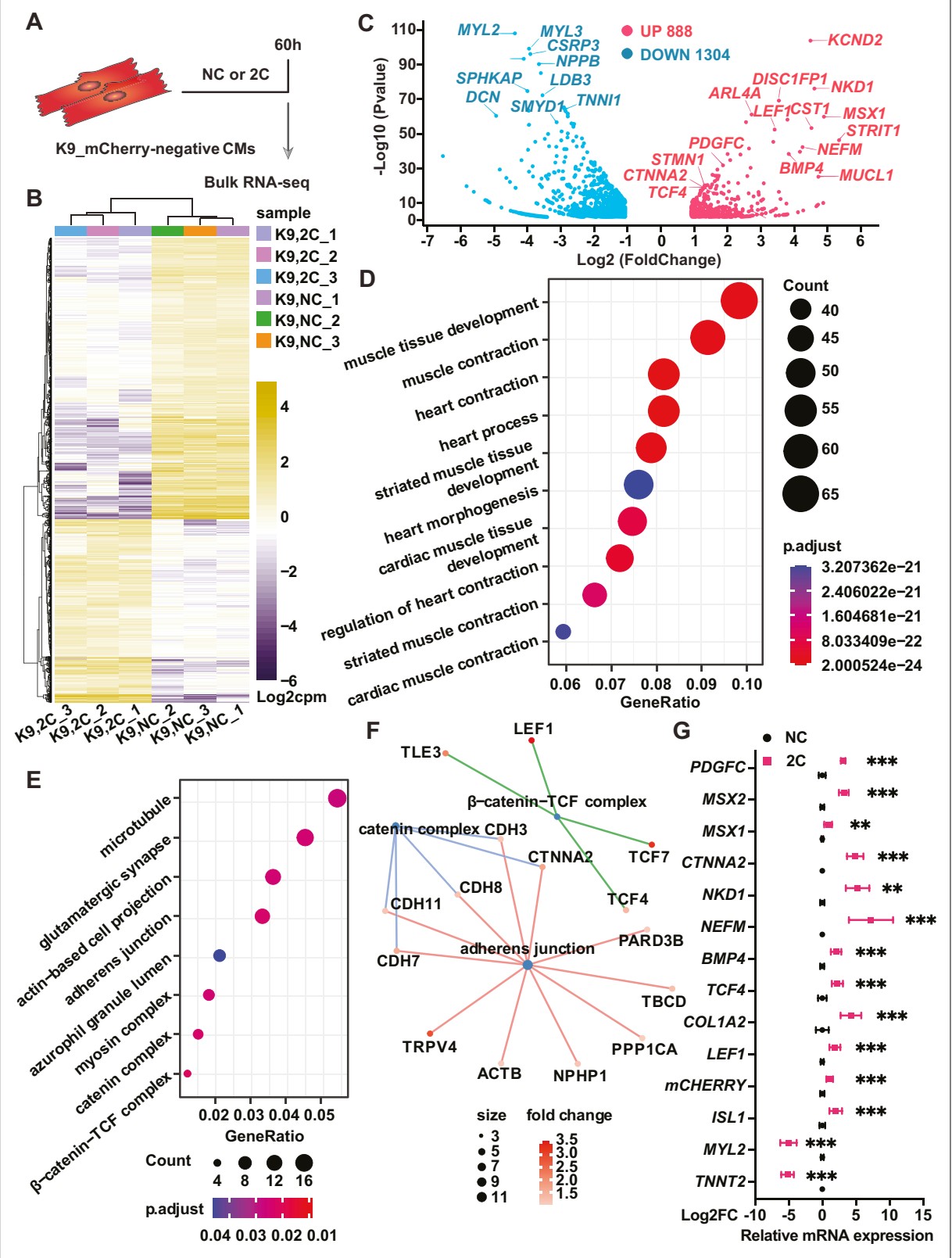

**Figure 5.** Bulk RNA-seq of analysis of 2C-treated ISL1/mCherry-negative cardiomyocytes (CMs). (**A**) Scheme of bulk RNA-seq analysis of K9-derived mCherry-negative CMs with DMSO (NC) or 2C treatment for 60 hr. (**B**) Heatmap of differentially expressed genes (DEGs) in ISL1/mCherry-negative CMs treated with DMSO (NC) or 2C for 60 hr. (**C**) Volcano plot showing genes significantly changed by 60 hr of 2C treatment. Gene ontology (GO) analysis of downregulated (**D**) and upregulated (**E**) genes in ISL1/mCherry-negative CMs by 2C treatment for 60 hr, compared to DMSO (NC) treated cells. (**F**)

*Figure 5 continued on next page*

*Figure 5 continued*

Plotting GO terms of upregulated genes by 2C treatment with cnetlpot. (**G**) Relative expression fold-changes of indicated genes in K9-derived ISL1/mCherry-negative CMs by 60 hr of DMSO (NC) or 2C treatment. Data are shown as mean ± SD. Multiple unpaired *t* tests. **p < 0.01, ***p < 0.001.

The online version of this article includes the following figure supplement(s) for figure 5:

**Figure supplement 1.** The individual and cooperative effects from CHIR99021 and A-485 on the induction of H9 human embryonic stem cell (hESC)-derived cardiomyocytes (CMs) into regenerative cardiac cells (RCCs).

condition. In comparison to 430 genes with H3K9Ac enrichment observed in DMSO-treated cells, which were mainly involved in cardiac muscle contraction (*Figure 7G, H*), 2C treatment markedly enhanced the enrichment of H3K9Ac in 123 genes associated with the transcription factor complex and Wnt signaling pathway (*Figure 7I, J*). In particular, 12 CM genes, including *TNNT2*, were ranked in the top 90 out of 430 annotated genes with H3K9Ac enrichment. In contrast, 7 RCC genes, including *ISL1*, were ranked in the top 50 out of 123 annotated genes with H3K9Ac enrichment.

Lastly, our findings were substantiated by DNA-binding motif analyses, which showed increased accessibility of ISL1-binding sites only in cells treated with A-485 or the 2C combination, underscoring the critical role of A-485 in activating RCCs genes through ISL1-dependent mechanisms (*Figure 7K*). Collectively, these results underscore the efficiency of combining CHIR99021 and A-485 in reprogramming CMs to RCCs, opening new avenues for developing therapies aimed at cardiac regeneration.

## Discussion

The generation of RCCs from CMs using the 2C treatment, a combination of CHIR99021 and A-485, represents a significant advancement in cardiac regenerative medicine. This study demonstrates that the synergistic interaction between these two molecules is essential for the reprogramming of endogenous CMs into RCCs, as evidenced by the absence of RCCs in postnatal rat hearts treated with either molecule alone (*Figure 4—figure supplement 2*). CHIR99021, a well-documented GSK3 inhibitor, activates the Wnt signaling pathway, thereby promoting the proliferation of human pluripotent stem cell-derived CMs (*Buikema et al., 2020*; *Quaife-Ryan et al., 2020*). It also induces H3K27Ac at promoters of critical cardiac genes such as *TNNT2*, *MYH6*, and *ISL1* (*Quaife-Ryan et al., 2020*). This observation aligns with our findings, which show a slight increase H3K27Ac levels in CMs following CHIR99021 treatment (*Figure 7D* and *Figure 7—figure supplement 1*). Moreover, our results reveal that the p300/CBP inhibitor A-485 facilitates CM dedifferentiation into RCCs by epigenetically suppressing CM-specific gene expression (*Figure 7D*), consistent with a decline in the fluorescence intensity of H3K27Ac and H3K9Ac in CMs (*Figure 7—figure supplement 1C, E*). Concurrently A-485 enhances modifications conducive to the RCC state, such as H3K9Ac and H3K27Ac on genes like ISL1 (*Figure 7D*). This dual mechanism of action underscores the potential of utilizing epigenetic modifiers like A-485 to influence cell fate decisions in terminally differentiated cells, suggesting broader applications in regenerative biology.

The RCCs induced by the 2C treatment not only re-expressed genes essential for embryonic cardiogenesis but also demonstrated the capability to differentiate into three cardiac lineages in vitro, mirroring the properties of ISL1[+] SHF cardiac progenitors (*Bu et al., 2009*; *Moretti et al., 2006*). Although these 2C-induced RCCs show limited proliferation compared to natural ISL1[+] progenitors, their potential for expansion under defined conditions presents an intriguing avenue for future research. These insights contribute significantly to the understanding of cardiac lineage plasticity and highlight the therapeutic potential of induced multipotent cardiovascular progenitors for heart repair and regeneration.

Despite these promising findings, several limitations warrant further investigation. The modest proliferation ability of 2C-induced RCCs relative to natural cardiac progenitors suggests that optimizing the culture conditions or treatment protocols may enhance their regenerative capacity. Additionally, the specific molecular mechanisms through which CHIR99021 and A-485 synergistically promote RCC formation remain incompletely understood. Future studies should aim to delineate these pathways more clearly, potentially through in vivo lineage tracing to map the fate decisions of individual cells. Moreover, the potential off-target effects of 2C treatment and its long-term impacts on cardiac function and structure need comprehensive evaluation. As we advance, exploring the scalability of this approach and its applicability to other types of terminally differentiated cells could

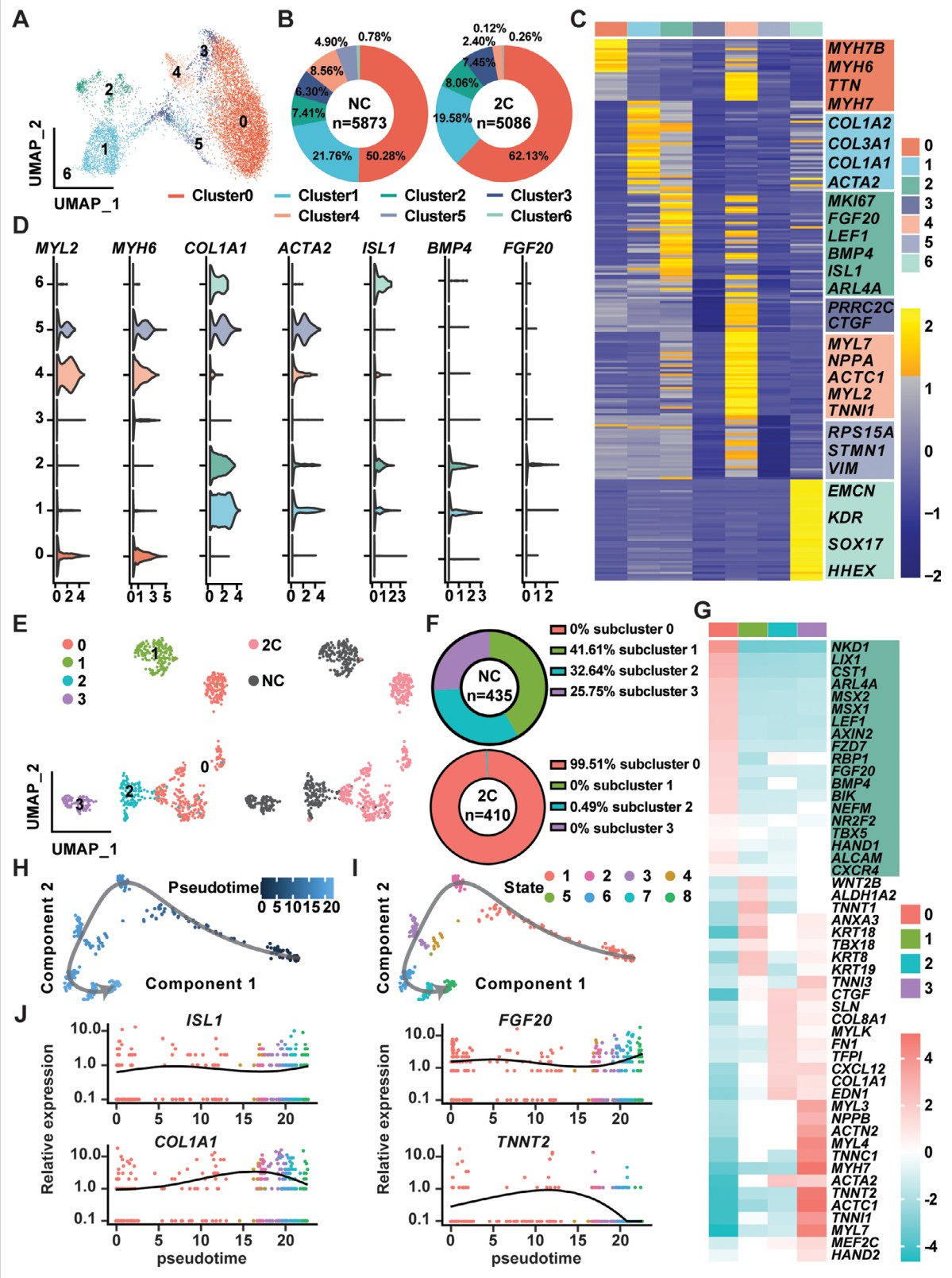

**Figure 6.** Single-cell RNA-seq of 2C-treated mCherry-negative cardiomyocytes (CMs). (**A**) UMAP analysis showing seven clusters in cells induced from K9-derived mCherry-negative CMs by treatment with DMSO (NC) and 2C for 60 hr. (**B**) The percentage of cells in the seven indicated clusters, following DMSO (NC) or 2C treatment. (**C**) Heatmap showing the differentially expressed genes in the cells from seven indicated clusters. The representative marker genes of seven indicated clusters were listed on the right. (**D**) Violin plots showing the expression levels of marker genes of CMs (*MYL2, MYH6*),

*Figure 6 continued on next page*

*Figure 6 continued*

intermediate cells (ICs) (*COL1A1*, *ACTA2*), and regenerative cardiac cells (RCCs) (*ISL1*, *BMP4*, *FGF20*) among cells from seven indicated clusters. (**E**) UMAP analysis showing the second-level clustering of cluster 2 into four subclusters (left), which exhibited dramatic distinction under condition of 2C or NC (right). (**F**) The percentage of cells in the four indicated subclusters within cluster 2, following DMSO (NC) or 2C treatment. (**G**) Heatmap showing the differentially expressed genes among cells from four subclusters of cluster 2. Genes related to RCCs are highlighted in the green blocks on the right. Pseudotime trajectory showing changes across various cell states upon 2C treatment, which were presented with different developmental pseudotime points (**H**) and cell states (**I**). (**J**) Curves showing the dynamic expression of representative genes of RCCs (*ISL1*, *FGF20*), ICs (*COL1A1*), and CMs (*TNNT2*) along indicated pseudotime points.

broaden the scope of regenerative therapies, offering new strategies for numerous degenerative conditions. In conclusion, this study not only sheds light on the cellular and molecular intricacies of cardiac cell dedifferentiation but also opens new pathways for the development of regenerative therapies aimed at heart disease.

## Materials and methods

### Mice

All animal experiments in this study were performed in accordance with the guidelines and regulations of IACUC (Institutional Animal Care and Use Committee) of Tsinghua University, Beijing, China. All the animal protocols used in this study have been approved by IACUC of Tsinghua University. All mice were housed in individually ventilated cages (maximal six mice per cage) at Tsinghua University. The mice/rats were maintained on a 12/12-hr light and dark cycle, and at 22–26°C with sterile pellet food and water ad libitum. The adult 129SvJ mice and pregnant SD1 rats at gestational day 17.5 were purchased from Beijing Vital River Laboratory Animal Technology. All adult mice were 8–12 weeks old when used for experiments.

### Drug administration

CHIR99021 (20 mg/kg, selleck) and A-485 (10 mg/kg, MCE) were administration in SD1 neonatal (at day of birth, P1) rats and 129SvJ adult mice by intraperitoneal (i.p.) injection as indicated. Administration with equal volume of DMSO was used as negative control.

### Chemically defined induction of hESCs differentiation into CMs

H9 and HUES7 hESCs (WiCell) were maintained in mTeSR1 medium (Stem Cell Technologies) on Matrigel (BD Biosciences) coated plates, according to the manufacturer's instructions. An established protocol (*Lian et al., 2013*; *Tohyama et al., 2013*) was modified for cardiac differentiation of hESCs. In brief, hESCs were seeded into the plates at 1.5 × 10⁵ cells/cm². When reaching 90–95% confluency, cells were cultured with RPMI/B27⁻ᴵᴺˢ media (the RPMI1640 basal medium (Gibco) with 1× B27 minus insulin supplement (Gibco)) containing 20 ng/ml hActivin A (R&D), 20 ng/ml hBMP4 (R&D), and 1.5 μM CHIR99021 (Selleck) for 2 days to initiate differentiation. Subsequently, the culture media were changed to RPMI/B27⁻ᴵᴺˢ media containing 5 μM IWP2 (Tocris) for 3 days and then changed to RPMI/B27⁻ᴵᴺˢ media for another 3 days. From day 8, the CM culture media RPMI/B27 was used to culture cells for 6 days, with changes every 2–3 days during this period. Contracting CMs were observed as early as day 10 of differentiation. To purify the CMs, cells at day 14 of differentiation were cultured in CM selection media (glucose-free DMEM (Gibco) supplemented with 4 mM lactate (Sigma)) for 4–6 days. CM selection media was changed every 2 or 3 days for eliminating dead cells. Cells were then digested by TrypLE (Gibco), reseeded onto fibronectin (BD Biosciences) coated plates with media containing MEM-a (Gibco), 5% fetal bovine serum (FBS; Gibco) and 5 μM Y-27632 (Selleck) for 24 hr and recovered in CM culture media for 4 days.

### Establish ISL1^mCherry/+ hESC lines

Knock-in of ISL1-mCherry reporter was performed in H9 and HUES7 hESC lines using CRISPR–Cas9 system (*Ran et al., 2013*). In details, three single-guide RNAs were designed and cloned into the PX459 plasmid (Addgene). The reporter gene mCherry was placed directly downstream of ISL1 start condon, and followed by the hygromycin resistance gene flanked by Loxp sites. All DNA fragments were contructed into the pEGFP-N3 vector (Addgene) from BamHI and NotI sites using NEBuilder

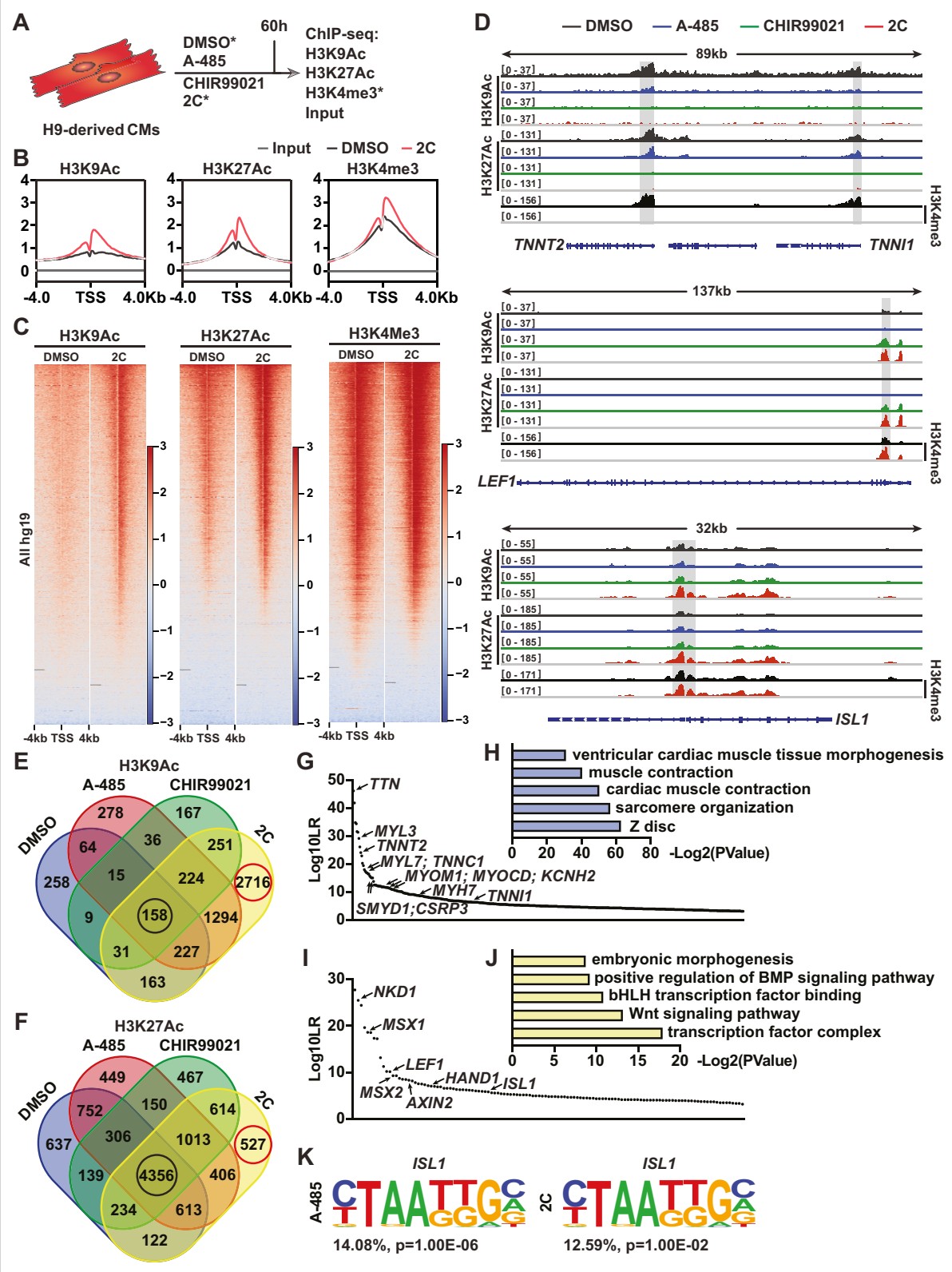

**Figure 7.** Chromatin immunoprecipitation-sequencing (ChIP-seq) analyses of chemical-treated H9 human embryonic stem cell (hESC)-derived cardiomyocytes (CMs). (**A**) Schematic illustration of ChIP-seq analysis of H9-derived CMs subjected to DMSO, A-485, CHIR99021, or 2C treatment for 60 hr. (**B**) Average ChIP-seq signal profiles showing the indicated histone modifications around the transcription start site (TSS) in the input and ChIP samples prepared from DMSO and 2C-treated cells. (**C**) Heatmap showing the whole-genome wide distribution of H3K9Ac, H3K27Ac, and H3K4me3

*Figure 7 continued on next page*

*Figure 7 continued*

peaks within a range of ±4 kb from TSSs in the cells treated with DMSO or 2C for 60 hr. (**D**) H3K9Ac and H3K27Ac peaks surrounding CM genes (*TNNT2* and *TNNI1*) and regenerative cardiac cell (RCC) genes (*LEF1* and *ISL1*) in the cells treated with DMSO, A-485, CHIR99021, or 2C for 60 hr, and H3K4me3 peaks surrounding the same genes in the cells treated by DMSO or 2C. The *y*-axis represents the number of counts. (**E, F**) Venn diagram showing the number of annotated genes with H3K9Ac or H3K27Ac enrichment in the cells treated with DMSO, A-485, CHIR99021, or 2C for 60 hr. Red circles indicate the number of genes with unique H3K9Ac or H3K27Ac enrichment induced by 2C treatment; black circles indicate the number of genes with H3K9Ac or H3K27Ac enrichment unaffected by any chemical treatment. The annotated genes with the most significant changes in H3K9Ac enrichment following treatment with DMSO (**G**) or 2C (**I**) were ranked by Log10LR and analyzed by gene ontologies (GOs) (**H, J**), respectively. (**K**) ISL1-binding motifs identified from the cells treated with A-485 or 2C.

The online version of this article includes the following figure supplement(s) for figure 7:

**Figure supplement 1.** H3K27Ac and H3K9Ac levels regulated by CHIR99021 and A-485 individually or in combination.

**Figure supplement 2.** Chemical treatment alters the expression of signature genes by regulating H3K27Ac and H3K9Ac in proximity.

**Figure supplement 3.** Chemical treatment differentially altered the peaks of H3K9Ac and H3K27Ac on cardiomyocyte (CM) and regenerative cardiac cell (RCC) genes.

HiFi DNA Assembly Master Mix (New England BioLabs) according to the manufacturer's instructions. Both CRISPR–Cas9 plasmids and trageting vectors were delivered into hESCs using P3 Primary Cell 4D-Nucleofector X Kit (Lonza) according to the manufacturer's instructions. Subsequenctly, cells with knock-in of ISL1-mCherry reporter were selected by hygromycin (500 ng/ml) (Thermo Fisher Scientific) and identified using PCR (Primer F: GGGCCCGGGGATCCGAAGGAAGAGGAAGA, Primer R: GAGGTCGA GATCCTAAGCTTGGC). To excise the selection cassette, transient expression of Cre recombinase was performed by transfection using Lipofectamine 3000 (Life Technologies), followed by flow cytometry-based sorting of EGFP-positive cells. The established ISL1$^{mCherry/+}$ hESC clones were genotyped by PCR (Primer F: CCGCGGGCCCGGGGATCCGTCA GTCCGCGGAGTCAAC, Primer R: TAGAGTCGCGGCCGCGCCGCAACCAACACATA) and sequencing.

## Chemical induction of RCCs from CMs

CMs were seeded onto culture well, fibronectin-coated 6-well-plates at a density of $1.6 \times 10^5$–$3.0 \times 10^5$ cells/cm$^2$ and cultured in CM culture media. When the CMs recovered contraction, the culture medium was changed with chemically reprogramming medium (MCDB131 Gibco), 25 mg/l NaHCO$_3$ (Sigma), 4 mM Glucose (Sigma), 1% Clutmine (Gibco), 1× N-2 supplement (Gibco), and 20% 1× B-27 supplement (Gibco), 10 µM CHIR99021 (Selleck), and 0.5 µM A-485 (MCE). Cells were induced to reprogram for 24, 48, or 60 hr, depending on experiments. Cells treated with equal volume of DMSO was used as negative control.

## Neonatal CMs isolation

Neonatal CMs (at postnatal day 3 or P3) were isolated following a previously reported protocol (*Sakurai et al., 2014*) with a slight modification. Briefly, the hearts were collected into 50 ml BD tubes containing 30 ml ice-chilled CMF-HBSS (Gibco) on ice. The rinsed hearts were transferred to a 10 cm plastic dish (CORNING) placed on ice and scissored after removal of large vessels and/or unwanted tissues. After adding frozen trypsin (Gibco) at a final concentration of 50 µg/ml, the dishes were sealed with parafilm and placed overnight at 4°C. On the next day, 5% FBS (Gibco) was added and incubated at 37°C for 30 min. After the addition of Leibovitz L-15 (Gibco) containing collagenase (Gibco), dishes were gently shaken for 45 min. When the tissue was homogenized, the cells were pipetted, filtered through 70 µm cell strainer (NEST), and transferred into a new 50 ml BD tube. Residual cells were resuspended with 5 ml of fresh Leibovitz L-15, filtered through 70 µm cell strainer, and transferred as well. Cells were further digested at 37°C for 40 min and then centrifuged at $200 \times g$ for 5 min. After re-suspension with 10 ml DMEM (Gibco) containing 10% FBS and 20 U/ml penicillin/streptomycin (Gibco), cells were transferred into 10 cm plastic dish and incubated at 37°C for 1 hr. Supernatant containing the CMs was collected and centrifuged at $200 \times g$ for 5 min. CMs were resuspended with DMEM containing 10% FBS and 10 U/ml penicillin/streptomycin, counted and seeded on laminin-coated 24-well plates at a density of $5 \times 10^5$ cells/cm$^2$. On the next day, the media was changed to DMEM/MEM (Gibco) containing 5% FBS, 10 U/ml penicillin/streptomycin and 0.1 mM BrdU (Sigma). After 48 hr, the media was changed to DMEM/MEM containing 5% FBS and 10 U/ml penicillin/streptomycin and the fresh media was changed every 2–3 until initiation of chemical treatment.

## MI model

MI was performed in adult mice at age of 9–12 weeks by LAD coronary artery ligation as previously described (*Mahmoud et al., 2014*). In details, adult mice were anesthetized by intraperitoneal injection of tribromoethanol (Sigma-Aldrich, 200 mg/kg) and artificially ventilated with tracheal intubation. Lateral thoracotomy at the third to fourth intercostal space was performed by blunt dissection of the intercostal muscles following skin incision. The LAD was ligated by a 7/0 non-absorbable silk suture. The successful ligation of the LAD was verified by visual inspection of the apex color turning to bloodless. Following ligation, the thoracic wall incisions and the skin wounds were sutured with a 4/0 non-absorbable silk suture. Mice were warmed for several minutes until recovery. In the sham controls, we performed the same procedures without LAD ligation.

## Small molecule libraries

To identify compounds that enable induction of ISL1-expressing cells from CMs, we performed a large-scale screening of a collection of 235 small molecules modulating signaling pathways, epigenetic modifications, metabolites and nuclear receptors involved in the cardiac development (*Supplementary file 1*), and two commercial compound libraries, Sigma-LOPAC (Sigma) and Selleck-FDA (Selleckchem). 5 µM or 0.5‰ DMSO was used as negative control in compound screening. Small molecules in our collection were purchased from Sigma, Tocris Bioscience, Selleck, and MedChemExpress.

## Immunocytochemistry and immunohistochemistry

For immunocytochemistry, cells were fixed with 4% paraformaldehyde (Sigma), blocked in phosphate-buffered saline (PBS; Gibco) containing 4% FBS (Gibco) and 0.3% Triton X-100 (Sigma) at room temperature for 30 min and incubated with the primary antibody diluted in PBS containing 5% skim milk (AMRESCO) and 0.1% Triton X-100 at 4°C overnight. After washing with PBS, cells were incubated with secondary antibodies conjugated with Alexa Fluor-488 or -549 (Life Technologies) diluted in PBS at room temperature for 1 hr and stained with 4',6-diamidino-2-phenylindole (Sigma) for 5 min. For immunohistochemistry, harvested hearts were fixed with 4% paraformaldehyde at 4°C for 1 hr and further dehydrated in 30% sucrose at 4°C overnight. Next, fixed hearts were embedded in optimal cutting temperature compound (SAKURA) and sectioned at 10 µm thickness with a cryostat (Leica CM1900). Immunostaining of cardiac tissue sections was performed following the procedure of immunocytochemistry described above. Quantitative analysis of the immunostained samples was captured and analyzed by Opera Phenix (PerkinElmer). The following antibodies were used: OCT4 (Santa Cruz Biotechnology; 1:400), SOX2 (Abcam; 1:400), NANOG (Abcam; 1:400), SSEA1 (Stemgent; 1:200), MESP1 (ASB, 1:200), MYL2 (Abcam; 1:500), mCherry (Abcam; 1:200), GATA4 (Abcam; 1:500), ISL1 (Developmental Studies Hybridoma Bank; 1:100), NKX2-5 (Santa Cruz Biotechnology; 1:200), TBX5 (Sigma, 1:200), SMA (Sigma, 1:500), TNNT2 (Sigma, 1:1000), GFP (Sigma, 1:500), CD31 (BD Biosciences; 1:200), VEcadherin (R&D; 1:200), MEF2C (Cell Signaling Technology; 1:200), NR2F2 (Cell Signaling Technology; 1:200), Phospho-Histone H3 (Cell Signaling Technology; 1:200), SSEA4 (R&D; 1:200), Smooth Muscle Actin-a (Thermo Scientific; 1:500). Images were captured using a confocal Zeiss LSM710 and Olympus IX83 inverted microscope. Trichrome staining was performed using the Trichrome Stain (Masson) Kit (Sigma, HT15).

## Flow-cytometry analysis and fluorescence-activated cell sorting

Cells subjected to flow-cytometry analysis were harvested and dissociated by using Versene (Gibco, used on D6) or TrypLE (Gibco, used on SD4). For direct flow-cytometry analysis of surface proteins, samples were incubated with Zombie VioletTM (eBioscience) diluted at 1:500 in PBS (Gibco) for 15 min in the dark. After washing once with PBS containing 2% FBS (Gibco), samples were stained with antibodies and analyzed using fluorescence-activated cell sorting (FACS). For analysis of intracellular proteins, cells were fixed and permeabilized with reagents in the Foxp3 Staining Buffer Set (eBioscience), blocked in PBS containing 2% FBS, and incubated with primary antibodies against ISL1 (Developmental Studies Hybridoma Bank; 1:200), TNNT2 (Sigma, 1:1000), mCherry (Abcam; 1:200). Isotype-matched normal IgGs (Life Technologies) served as negative controls. After stained with secondary antibodies conjugated with Alexa Fluor-488 or -549 (Life Technologies), cells were analyzed and quantified by flow cytometry using a BD FACSAria III Cell Sorter (BD Biosciences). To sort K7- or K9-derived mCherry$^+$ cells, mCherry-negative CMs or TNNT2-EGFP$^+$ CMs in this study,

corresponding living cells were harvested and incubated with Zombie VioletTM (eBioscience) at 1:500 in PBS for 15 min in the dark. After washing with PBS containing 2% FBS, cells were resuspended with CM culture media and subject to FACS using the BD FACSAria III Cell Sorter.

## RNA extraction, reverse-transcription PCR and quantitative real-time PCR

Total RNA extracted from $>10^5$ cells with AxyPrep Multisource RNA Miniprep Kit (AXYGEN Biosciences), according to the manufacturer's instructions, or extracted from $<10^5$ cells using TRIzol (Gibco). After elimination of genomic DNA by DNaseI, cDNA was synthesized by reverse transcription of 1 µg total RNA with iScript cDNA Synthesis Kits (Bio-Rad). Quantitative RT-PCR was performed and analyzed with iQ SYBR Green Supermix (Bio-Rad) using CFX384 Touch Real-Time PCR Detection System (Bio-Rad). Relative mRNA expression of specific genes was normalized to the level of human *GAPDH* transcripts.

## Western blot

Cells were lysed on ice in RIPA buffer (50 mM Tris–HCl pH 7.5, 150 mM sodium chloride, 0.25% sodium deoxycholate, 0.1% Nonidet P-40, and 0.1% Triton X-100) containing protease and phosphatase inhibitors (Roche). The homogenized samples were centrifuged at 14,000 rpm for 5 min at 4°C, and the supernatant of each sample was transferred into a new respective tube and quantified by Bradford Assay. After addition of 5× sodium dodecyl sulfate (SDS) sample buffer (50 mM Tris–HCl PH 7.5, 150 mM NaCl, 0.25% Sodium deoxycholate, 0.1% Nonidet P-40, 0.1% Triton X-100, 5% β-mercaptoethanol), samples were boiled for 5 min. Next, 10 µg of total protein from each sample was subjected to SDS–polyacrylamide gel electrophoresis (SDS–PAGE) to separate proteins based on their different molecular weight. Western blot was performed using antibodies of β-actin (Santa Cruz Biotechnology; 1:500), and ISL1 (Developmental Studies Hybridoma Bank; 1:200). The results were visualized by AI600 (GE), and quantitatively analyzed by ImageJ.

## Lentiviral transduction

All lentiviral shuttle plasmids were constructed with NEBuilder HiFi DNA Assembly Master Mix (New England BioLabs) according to the manufacturer's instructions. Briefly, DNA fragments with ~20 bp overlapping ends were generated by restriction enzyme digestion or PCR. Then DNA fragments were mixed with the Assembly Master Mix at a volume ratio of 1:1 and incubated at 50°C for ~30 min. The products were transformed into Trans1-T1 Phage Resistant Chemically Competent Cell (TransGen Biotech). The TNNT2 promoter was amplified from genomic DNA by PCR (Primer F: GTCATGGA GAAGACCCACCTT, Primer R: GATCCTGGAGGCGTCTGC). All of the plasmids used in this study were purchased from Addgene. After package of lentivirus in HEK293T cells, the concentrated lentiviral supernatants were mixed with fresh CM culture media at a volume ratio of 1:1 and used to transduce isolated CMs for 48 hr. Next, CMs were cultured with CM culture media containing 2 µM Tamoxifen (Selleck) for another 6 days. The EGFP⁺ CMs sorted by FACS were used in 2C-induced reprogramming.

## Multibarcode RNAseq

K9-derived mCherry-negative CMs sorted by FACS were treated with DMSO or 2C for 60 hr and then prepared for Smart-Seq2. The RNA-seq libraries were generated from these samples according to a previously reported Smart-Seq2 protocol with minor modifications (*Picelli et al., 2013*). Briefly, cells were lysed in lysis buffer containing RNase inhibitor (TaKaRa). RNAs were captured with 25 nt oligo (dT) primers and reversed into cDNAs. After amplification and purification, cDNAs were sheared to approximately 300 bp by CovarisS2 and captured by DynabeadsR MyOne Streptavidin C1 beads (Thermo Fisher). All libraries were constructed using a Kapa Hyper Prep Kit (Kapa Bio-systems) and sequenced on 150 bp paired-ends Illumina Novaseq 6000 platform.

## scRNA-seq analysis

After sorted by FACS, K9-derived mCherry-negative cells were treated with DMSO or 2C for 60 hr, and then subject to scRNA-seq. The single-cell suspensions were prepared in PBS containing 0.04% bovine serum albumin (Sigma). The scRNA-seq libraries were constructed by Chromium Next

GEM Single Cell 3′ Kit v3.1 up at Chromium Controller (10× Genomics) and the quality of these libraries were assessed by Qubit 4.0 and the Agilent 2100. Sequencing was performed on the Illumina NovaSeq 6000 with paired-end reads. Each sample was aligned to the human genome GRCh38, and analyzed using the 10× cell ranger 6.0.0, which employed the STAR sequence aligner (*Dobin et al., 2013*). Cell barcodes were filtered using cellranger 3.0.2, and then downstream analysis was conducted using Seurat (v4.0.1). We integrated two samples using variable genes, and performed downstream analysis. Briefly, low-quality cell data were firstly removed and gene expression matrices were then normalized using the LogNormalized method. Subsequently, we clustered cells using the FindClusters functions and performed non-linear dimensional reduction with the RunUMAP function. Marker genes for each cluster were identified by the 'bimod' (Likelihood-ratio test) with default parameters via the FindAllMarkers function in Seurat (v4.0.1). GO analysis was performed using Enrichr (*Kuleshov et al., 2016*). Pseudotime trajectories analysis was conducted using the Monocle 2 (v2.18.0) (https://github.com/cole-trapnell-lab/monocle-release, *Qiu and Trapnell, 2019*). Sequencing data have been deposited in the Sequence Read Archive of the NCBI under the BioProject accession number PRJNA903530.

## Chromatin immunoprecipitation sequencing

The chromatin immunoprecipitation (ChIP) assay was executed with the SimpleChIP Plus Enzymatic Chromatin IP Kit (Magnetic Beads) provided by Cell Signaling Technology (#9005) following the manufacturer's instructions. Briefly, approximately $4 \times 10^6$ cells (equivalent to a 15-cm culture dish) were utilized for each ChIP assay. Initially, the cells were subjected to fixation with 1% formaldehyde for 10 min at room temperature. The fixation process was halted by the addition of 2 ml of 10× glycine for 5 min. The subsequent steps involved scraping the cells, lysing, digesting, and shearing them by sonication. The lysates were clarified by centrifugation, and 2% of the supernatant was transferred as an 'Input Sample', which could be preserved at −20°C for future use. The remaining lysates supernatant was incubated overnight at 4°C with the immunoprecipitating antibody. This was followed by adding 30 µl of Protein G Magnetic Beads to each IP reaction and incubating them for 2 hr at 4°C with rotation. The beads were then washed, and the chromatin was eluted from the antibody/protein G magnetic beads complex. To all samples, including the 2% input sample from previous steps, 6 µl of 5 M NaCl and 2 µl of Proteinase K were added to reverse cross-links. This was followed by a 2-hr incubation at 65°C and DNA purification using Spin Columns. Subsequently, these immuno-enriched DNA samples were prepared for Next Generation Sequencing (NGS) after constructing a DNA library with the NEBNext Ultra II DNA Library Prep Kit (NEB, USA, Catalog #: E7645L). The qualified libraries were pooled and sequenced on Illumina platforms using a PE150 strategy at Novogene Bioinformatics Technology Co., Ltd (Beijing, China). Finally, the raw reads were subjected to quality control using Fastp and then aligned to the hg19 genome with hisat2. Peak calling was conducted using macs2 and deeptools, and the resultant plots were generated using the IGV software, and the peaks were not normalized by sequencing depth.

## Statistical analysis

Values are presented as means ± SD. Each figure legend has detailed information explaining the statical test used to show significance. All statistical tests were performed on GraphPad Prism 10, and $p < 0.05$ was considered statistically significant (ns, $p > 0.05$; *$p < 0.05$; **$p < 0.01$; ***$p < 0.001$; ****$p < 0.0001$). Microscopy images were selected form the total quantified images. Microscopy quantified fields were whole fields of well in cell culture plates, and were randomly acquired in immunostained sections.

## Acknowledgements

This work is supported by the National Natural Science Foundation of China (32030031 to SD), Beijing Natural Science Foundation (JQ22016 to TM), the National Key R&D Program of China (2022YFA1103704 to SD; 2022YFA1104503 to YN), Center for Life Sciences (to SD). We also thank the Center for Pharmaceutical Technology, Tsinghua University for the activity screening platform, Biomarker Technologies Corporation, Beijing, China and BeiJing Geek Gene Technology Co Ltd for technical support, and support from Tsinghua-Peking Center for Life Sciences.

# Additional information

## Funding

| Funder | Grant reference number | Author |
|---|---|---|
| National Natural Science Foundation of China | 32030031 | Sheng Ding |
| National Key Research and Development Program of China | 2022YFA1103704 | Sheng Ding |
| National Key Research and Development Program of China | 2022YFA1104503 | Yu Nie |
| Beijing Natural Science Foundation | JQ22016 | Tianhua Ma |
| Center for Life Sciences | | Sheng Ding |

The funders had no role in study design, data collection, and interpretation, or the decision to submit the work for publication.

## Author contributions

Wei Zhou, Conceptualization, Resources, Data curation, Software, Formal analysis, Supervision, Validation, Investigation, Visualization, Methodology, Writing – original draft, Project administration, Writing – review and editing; Kezhang He, Data curation, Software; Chiyin Wang, Data curation; Pengqi Wang, Data curation, Methodology, Writing – original draft; Dan Wang, Data curation, Formal analysis, Writing – original draft; Bowen Wang, Conceptualization, Data curation, Software, Writing – original draft; Han Geng, Writing – original draft; Hong Lian, Methodology, Writing – original draft; Tianhua Ma, Yu Nie, Conceptualization, Writing – original draft, Writing – review and editing; Sheng Ding, Conceptualization, Supervision, Writing – original draft, Writing – review and editing

## Author ORCIDs

Wei Zhou ⓘ https://orcid.org/0000-0003-1069-1151
Chiyin Wang ⓘ https://orcid.org/0000-0001-6128-8587
Tianhua Ma ⓘ https://orcid.org/0000-0002-4707-263X
Sheng Ding ⓘ https://orcid.org/0000-0002-3354-1263

## Ethics

All animal experiments in this study were performed in accordance with the guidelines and regulations of IACUC (Institutional Animal Care and Use Committee) of Tsinghua University, Beijing, China. All the animal protocols used in this study have been approved by IACUC of Tsinghua University. All mice were housed in individually ventilated cages (maximal six mice per cage) at Tsinghua University. The mice/rats were maintained on a 12/12-hr light and dark cycle, and at 22–26 °C with sterile pellet food and water ad libitum. The adult 129SvJ mice and pregnant SD1 rats at gestational day 17.5 were purchased from Beijing Vital River Laboratory Animal Technology. All adult mice were 8–12 weeks old when used for experiments.

Reviewer #1 (Public review): https://doi.org/10.7554/eLife.93405.3.sa1
Reviewer #2 (Public review): https://doi.org/10.7554/eLife.93405.3.sa2
Reviewer #3 (Public review): https://doi.org/10.7554/eLife.93405.3.sa3
Author response https://doi.org/10.7554/eLife.93405.3.sa4

# Additional files

## Supplementary files

• Supplementary file 1. Table of compound library collected based on hypotheses. The compounds from the lab's proprietary library, along with their respective targets and working concentrations, are provided in the table.

• MDAR checklist

## Data availability

Sequencing data have been deposited in the Sequence Read Archive of the NCBI under the BioProject accession number PRJNA903530. All data generated or analysed during this study are included in the manuscript and supporting files; source data files have been provided for *Figure 1* and *Figure 3—figure supplement 1*. *Figure 1—source data 2* contain the original western blots for *Figure 1F*; *Figure 3—figure supplement 1—source data 2* contain the original gel graph for *Figure 3—figure supplement 1*.

The following dataset was generated:

| Author(s) | Year | Dataset title | Dataset URL | Database and Identifier |
|---|---|---|---|---|
| Zhou W, He K, Wang C, Wang P, Wang D, Wang B, Geng H, Lian H, Ma T, Nie Y, Ding S | 2024 | Homo sapiens raw sequence reads | https://www.ncbi.nlm.nih.gov/bioproject/PRJNA903530 | NCBI BioProject, PRJNA903530 |

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
