## [Editor Report · eLife Assessment]

This manuscript offers **valuable** information on the combinatory effect of small molecules, CHIR99021 and A-485 (2C), during the reprogramming of mature cardiomyocytes into regenerative cardiac cells on stimulating cardiac cell regeneration. Although the study used several hESC lines and an in vivo model of myocardial injury to demonstrate the regenerative potential of cardiac cells, the manuscript is still **incomplete** as several concerns remain unanswered, including the lack of validation of the conclusions from scRNA-seq. It is still unclear how a small fraction of dedifferentiating cardiac cells can offer such broad effects on regeneration both in vitro and in vivo. If validated, this study might unlock potential therapeutic strategies for cardiac regeneration.

---

## [Referee Report · Reviewer #1 (Public review)]

The present manuscript by Zhou and colleagues investigates the impact of a new combination of compounds termed CHIR99021 and A-485 on stimulating cardiac cell regeneration. This manuscript fits the journal and addresses an important contribution to scientific knowledge.

Comments on latest version:

The authors have addressed all of our comments.

---

## [Referee Report · Reviewer #2 (Public review)]

Summary:

This manuscript reports that a combination of two small molecules, 2C (CHIR99027 and A-485) enabled to induce the dedifferentiation of hESC-derived cardiomyocytes (CMs) into regenerative cardiac cells (RCC). These RCCs had disassembled sarcomeric structures and elevated expression of embryonic cardiogenic genes such as ISL1, which exhibited proliferative potential and were able to differentiate into cardiomyocytes, endothelial cells, and smooth muscle cells. Lineage tracing further suggested that RCCs originated from TNNT2+ cells, not pre-existing ISL1+ cells. Furthermore, 2C treatment increased the numbers of RCC cells in neonatal rat and adult mouse hearts, and improves cardiac function post-MI in adult mice. Mechanistically, bulk RNA-seq analysis revealed that 2C led to elevated expression of embryonic cardiogenic genes while down-regulation of CM-specific genes. Single-cell RNA-seq data showed that 2C promoted cardiomyocyte transition into an intermediate state that are marked with ACTA2 and COL1A1, which subsequently transform into RCCs. Finally, ChIP-seq analysis demonstrated that CHIR99027 enhanced H3K9Ac and H3K27Ac modifications in embryonic cardiac genes, while A-485 inhibited these modifications in cardiac-specific genes. These combined alterations effectively induced the dedifferentiation of cardiomyocytes into RCCs. Overall, this is an important work, presenting a putative cardiac regenerative cell types that may represent endogenous cardiac regeneration in regenerative animals. With that said, here are suggestions for the authors:

Strengths:

Overall, this work is quite comprehensive and is logically and rigorously designed. The phenotypic and functional data on 2C are strong.

Weaknesses or suggestions:

(1) In Figure 4, the authors should perform additional experiments on analyzing 2C effect on cardiomyocytes, endothelial cells, and fibroblasts in adult mouse hearts after myocardial infarction.

(2) In Figures 5-7, the mechanistic insights of 2C are primarily derived from transcriptomic and genomic datasets without experimental verification.

(3) The authors should compare transcriptomic profiling of the RCCs with other putative cardiac progenitors from public databases.

---

## [Referee Report · Reviewer #3 (Public review)]

Summary:

The ability of cardiac cells to regenerate has been the object of intense (and sometimes controversial) research in biology. While lower organisms can robustly undergo cardiac regeneration by reactivation of embryonic cardiogenic pathway, this ability is strongly reduced in mice, both temporally and qualitatively. Finding a way to derive precursor cells with regenerative ability from differentiated cells in mammals has been challenging.

Zhou, He and colleagues hypothesized that ISL-1-positive cells would show regenerative capacity and developed a small molecules screen to dedifferentiate cardiomyocytes (CM) to ISL1-positive precursor cells. Using hESC-derived CM, authors found that the combination of both, WNT activation (CHIR99021) and p300 acetyltransferase inhibition (A-485) (named 2C protocol) induces CM dedifferentiation to regenerative cardiac cells (RCCs). RCCs are proliferative and re-express embryonic cardiogenic genes while decreasing expression of more mature cardiac genes, bringing them towards a more precursor-like state. RCCs were able to differentiate to CM, smooth muscle cells and endothelial cells, highlighting their multipotent property. In vivo administration of 2C in rats and mice had protective effects upon myocardial infarction.

Mechanistically, authors report that 2C protocol drives CM-specific transcriptional and epigenetic changes.

Strengths:

The authors made a great effort to validate their data using orthogonal ways, and several hESC lines. The use of lineage tracing convincingly showed a dedifferentiation from CM. They translate their findings into an in vivo model of myocardial injury, and show functional cardiac regeneration post injury. They also showed that 2C could surprisingly be used as preventive treatment. Together their data may suggest a regenerative effect of 2C both in vitro and in vivo settings. If confirmed, this study might unlock therapeutic strategy for cardiac regeneration.

Weaknesses:

Updated General comments:

Experimental design & Interpretation

(1) The titration provided by the author following the first round of revision is puzzling to me. Based on the authors explanation, the initial screen was performed using 10uM of A-485, allowing the authors to choose CHIR + A-485 as a combination of drugs increasing Isl1-positive cells. However, in the titration provided, the combination of CHIR + 10uM of A-485 (used during the screen) shows *no* increase of the percentage of Isl-1-positive cells compared to DMSO control. How is that possible? Can the authors provide a transparent explanation of the experimental design for their screen. How was A-485 isolated from the 4000+ compounds tested if it does not show any effect on the titration? This titration raises significant concerns about the rational of following up with the combination of compounds.

(2) The authors have not really addressed the concern raised earlier. If only ~1% of the cells de-differentiate and become Isl-positive, how can anybody quantify a nuclear/cytosolic ratio at the global population and show statistical significant when only 1% of the cells should be different?

(3) Authors now provide a quantification of the effect of I-BET-762 (Supp 1H). While the authors state " [the combination of CHIR + I-BET-762] was less effective than A-485 in combination with CHIR99021", the figure provided does not test that. A side-by-side comparaison of the effect of A485 and I-BET should have been performed on the same graph. I-BET increases by 4 fold, while A-485 increases by 5-fold, which, based on the variation of their data, will unlikely be statistically different. The rational for disregarding the effect of I-BET-762 is therefore weakened.

(4) Why NR2F2 is statistically significant in one set of experiments (Fig 2 - Fig. supplement 1) and then non-significant in another set (Fig. 1G) using the exact same experiment design (NC vs 2C for 60h) and similar statistical test applied?

Statistics & Data Acquisition

(1) Authors should refrain from deriving statistics from 2 biological repeats (Figure 3G).

(2) Authors still do not state whether the normality of their data was tested.

(3) What is the rational for using a two-way ANOVA for Fig 3G? Authors are only comparing the effect of their treatment for each marker. Same question for most panels from Figure 1, Fig 2C, 2F, and throughout the manuscript. This needs clarification/justification especially because in other experiments, they used multiple unpaired t-test (Fig 2 - Fig. supplement 1).

Others

(1) Authors should try to make their manuscript colorblind-friendly: No modification added following this comment.

---

## [Author Response]

The following is the authors’ response to the original reviews.

**Reviewer 1**
Overall, this work is quite comprehensive and is logically and rigorously designed. The phenotypic and functional data on 2C are strong.

Thank you for your positive feedback on our findings!

(1) Comment from Reviewer 1 suggesting the mechanistic insights of 2C are primarily derived from transcriptomic and genomic datasets without experimental verification.

Thank you for emphasizing the importance of experimental validation to support our transcriptomic and genomic findings. We acknowledge the gap in direct experimental evidence for the mechanistic insights of section 2C and recognize the value of such validation in strengthening our conclusions. While we recognize the importance of such validation, our current dataset lacks the comprehensive preliminary results necessary for inclusion in the supplemental material. We believe that the mechanistic insights presented offer a substantial foundation for the future research, where we aim to explore these aspects in depth with targeted experimental approaches.

**Reviewer 2**
Together their data may suggest a regenerative effect of 2C both in vitro and in vivo settings. If confirmed, this study might unlock therapeutic strategy for cardiac regeneration.

Thank you for your positive comment on the significance of our findings and the valuable therapeutic potential of 2C in cardiac regeneration!

(1) Comment from Reviewer 2 pointing out the the main hypothesis (line 50) that Isl1 cells have regenerative properties is not extremely novel.

We agree with the reviewer that Isl1-positive cells possess regenerative properties. Following the reviewer’s suggestion, we have revised the original wording (line 46 in the revised manuscript).

(2) Comment from Reviewer 2 asking for providing a rationale for this 20x reduction of A-485 concentration? It would be useful to get a titration of this compound for the effects tested.

As suggested by the reviewer, we have added the titration results of A-485 in Figure 1—figure supplement 1F-G.

(3) Comment from Reviewer 2 confusing to clearly understand what proportion of CMs dedifferentiate to become RCCs. The lineage tracing data suggests only 0.6%-1.5% of cells undergo this transition. It is difficult to understand how such a small fraction can have wide effects in their different experimental settings. This is specifically true when the author quantified nuclear and cytosolic area on brightfield pictures - would the same effect on nuclear/cytosolic area be observed in Isl1 KO cells.

We appreciate the reviewer's insightful observation on the proportion of CMs undergoing dedifferentiation into RCCs and the potential impact of this subset on our experimental outcomes. The lineage tracing data indicating that only 0.6%-1.5% of CMs transition to RCCs indeed reflects a modest proportion. This observation raises valid questions regarding the broader implications of such a limited fraction in the context of cardiac regeneration and the experimental effects reported. It's important to note that while the proportion of CMs dedifferentiating into RCCs is small, the biological significance and potential impact of these RCCs could be disproportionately large. Emerging evidence suggests that even a minimal number of stem or progenitor cells can exert significant effects on tissue repair and regeneration, possibly through paracrine mechanisms or by acting as key signaling centers within the tissue microenvironment (Fernandes et al., 2015). Regarding the specific question about 2C’s effects on nuclear/cytosolic area in Isl1 knockout (KO) cells, we appreciate the suggestion and consider that such comparative studies would provide valuable insights for future comprehensively understanding the significant impact of 2C-induced RCCs in future search. In addition, ISL1 KO cells are also described in detail in the article published in *eLife* in 2018 by Quaranta et al.

(4) Comment from Reviewer 2 asking for the effect of CHIR + I-BET-762 alone.

As suggested by the reviewer, we have added the results of CHIR + T-BET-762 in Figure 1—figure supplement 1H.

(5) Comment from Reviewer 2 suggesting a transparent explaination about the effects of A-485 on acetylation status.

We thank the reviewer for highlighting the confusion regarding the effects of A-485 on the acetylation status of H3K27Ac and H3K9Ac. Upon re-examination of our data and statements, we recognize the need for clarity in our explanation and the inconsistency it may have caused (lines 223-231 on page 8).

Initially, our observations suggested a selective effect of A-485 on H3K27Ac based on early experimental results (Figure 7—figure supplement 1). This conclusion was drawn from preliminary analyses that focused predominantly on this specific histone mark. However, upon further comprehensive examination of our data, including additional replicates and more sensitive detection methods, we observed that A-485 also impacts H3K9Ac levels (Figure 7—figure supplement 1F). This latter finding emerged from expanded datasets that were not initially considered in our preliminary conclusions.

The "further analyses" mentioned referred to these subsequent experimental investigations, which included chromatin immunoprecipitation (ChIP) assays and extended sample sizes, providing a more robust dataset for evaluating the effects of A-485. We understand the importance of transparency and rigor in scientific communication. To address this, we have revised the manuscript to clearly delineate the progression of our analyses and the evidential basis for our revised understanding of A-485's effects. This includes a detailed description of the methodologies employed in our follow-up experiments (line 537 on page 27), the statistical approaches for data analysis (lines 226-227 in supporting information), and how these led to the updated interpretation regarding A-485's impact on histone acetylation (lines232-269).

(6) Comment from Reviewer 2 asking for the difference in the ChIP peaks representation of the y-axis on the ChIP traces.

Thank you for raising this quest. Actually, we did not normalise the sequencing depth and the y-axis represents the number of counts (line 537 on page 27 and lines 226-227 in supporting information).

(7) Comment from Reviewer 2 suggesting the possibility of testing this 2C protocol on mESCs to see if similar changes are subject to and how these mouse RCCs differ transcriptionally from Isl1+ progenitor cells isolated from neonatal mice (P1-P5)?

Thank you for your insightful questions. Testing the 2C protocol on mouse embryonic stem cells (mESCs) to observe if similar changes occur presents an excellent opportunity to further validate the versatility and applicability of our findings across different stem cell models. We agree that such experiments would not only strengthen the current study but also provide valuable insights into the conservation of mechanisms across species. We are currently in the process of setting up experiments to address this very question and anticipate that the results will significantly contribute to our understanding of cardiomyocyte differentiation processes. Regarding the transcriptional comparison between mouse regenerative cardiac cells (RCCs) induced by our 2C protocol and Isl1+ progenitors isolated from neonatal mice (P1-P5), this comparison is indeed crucial for delineating the specific identity and developmental potential of the RCCs generated. However, a comprehensive side-by-side transcriptomic analysis is required to systematically identify these differences and understand their biological implications. We plan to undertake this analysis as part of our future studies, which will include detailed RNA sequencing and comparative gene expression profiling to elucidate the transcriptional similarities and differences between these cell populations. These future directions will enhance our current findings, provide a deeper mechanistic understanding, and confirm the potential of the 2C protocol in regenerative medicine applications. We appreciate the reviewer's suggestions and acknowledge the importance of these experiments in advancing the field.

(8) Comment from Reviewer 2 with a suggestion to have a precise clarification of statistics & data acquisition.

As suggested by the reviewer, we have revised clarifications to make them clearer (lines 228-233 in supporting information and a precise description of each paragraph involving statistical analyses).

**Reviewer 3**
The findings may have a translation potential. The idea of promoting the regenerative capacity of the heart by reprogramming CMs into RCCs is interesting.

Thank you for your appreciation of the significance and translational potential of our findings!

(1) Comment from Reviewer 3 suggesting the mechanism involved in the 2C-mediated generation of RCCs is unclear and the lead found in the RAN-seq and ChIP-seq are not experimatally validated.

We acknowledge the reviewer's concern regarding the lack of experimental validation for the mechanisms identified through RNA-seq and ChIP-seq analyses in the generation of RCCs from the 2C state. We understand the importance of substantiating these molecular leads with empirical data to strengthen our conclusions. Currently, our findings are based on in-depth bioinformatic analyses, which have provided us with valuable insights and a strong basis for hypothesis generation. Moving forward, we plan to prioritize experimental validation of key pathways and targets identified in our study. This will include designing targeted experiments to elucidate the functional roles of these mechanisms in the 2C-mediated generation of RCCs. We appreciate the opportunity to clarify our approach and future directions, and we are committed to addressing this gap in subsequent work.

(2) Comment from Reviewer 3 considering the very low number of RCCs (0.6%-1.5% of cells) generated cannot protect the heart from MI, and whether 2C affects the the survival or metabolism of existing CM under hypoxia conditions, and what percentage of cells are regenerated by 2C treatment post-MI?

We appreciate the reviewer's insightful queries regarding the protective effects of 2C treatment against myocardial infarction (MI) given the low percentage of RCCs generated. It is our hypothesis that the benefits of 2C treatment extend beyond mere cell numbers. We propose that 2C may enhance the survival and metabolic resilience of existing CMs under hypoxic conditions, thereby contributing to cardiac protection post-MI. Our future investigations will aim to quantify the precise percentage of cells regenerated by 2C treatment post-MI and explore its broader impacts on cardiac tissue survival and repair mechanisms.

(3) Comment from Reviewer 3 suggesting the administration of 2C in mice, as well as whether 2C affects cardiac function under basal conditions and any physiology in mice, and the need to examine cardiac structural and functional parameters after administration of 2C.

We appreciate the reviewer's interest in the potential effects of 2C administration on cardiac function and overall physiology in mice. While we observed a decrease in body weight at P5 compared to controls, our immunofluorescence staining did not indicate any changes in cardiac structure (Figure 4— figure supplement 1E). This suggests that while 2C administration impacts neonatal rat physiology, it does not adversely affect cardiac structure under basal conditions. Further investigations are planned to assess the functional parameters of the heart post-2C administration to comprehensively understand its effects.

(4) Comment from Reviewer 3 suggesting the potential effects of 2C on other cell types of the heart, including fibroblasts and endothelial cells, in vitro and in vivo.

We value the reviewer's suggestion to explore the effects of 2C on various cardiac cell types, including fibroblasts and endothelial cells, both in vitro and in vivo. We acknowledge the importance of understanding the broader impact of 2C treatment across different cell populations within the heart, given its potential protective effects. To address this, we are designing a series of experiments to assess 2C's influence on these cell types, aiming to elucidate any changes in their behavior, proliferation, and function following treatment. This comprehensive approach will allow us to better understand the mechanistic basis of 2C's cardioprotective effects.

(5) Comment from Reviewer 3 suggesting validation the effect of 2C in a dose-dependent manner.

As suggested by the reviewer, we have supplemented the effect of 2C in dose-dependent (Figure 1— figure supplement 1F-G).

(6) Comment from Reviewer 3 suggesting an explanation of how A-485 affects H3K27Ac and H3K9Ac.

We appreciate the reviewer pointing out the discrepancy regarding the effects of A-485 on H3K27Ac and H3K9Ac. Upon re-examination of our data, we realize that our initial interpretation may have overlooked the broader impact of A-485 on histone acetylation patterns. It appears that A-485 does indeed influence both H3K27Ac and H3K9Ac, contrary to our initial statement. This oversight will be corrected in our revised manuscript, where we will provide a more detailed analysis and discussion of A-485's impact on these histone marks, alongside an explanation for the observed effects (lines 223-269 across page 8-9).

(7) Comment from Reviewer 3 with a correction to use "regeneration" at the screeing stage.

As suggested by the reviewer, we have amended the wording in the text (line 66 on page 3).

**Reviewer 4**
Comment from Reviewer 4 suggesting more information that clarifies and justifies the hypothesis.

As suggested by the reviewer, we added more information to clarify and justify the hypothesis (lines 39-47 on page 3).

(1) Comment from Reviewer 4 pointing out the story line is not well developed.

To address the reviewer’s question, we revised the manuscript to ensure a smooth and coherent logical flow.

(2) Comment from Reviewer 4 pointing out the purpose in choosing to study ISL1-CMs.

As raised by the reviewer, we have clarified the rationale for using ISL1 as a marker to define RCCs in revised manuscript (lines 39-47 on page 3).

(3) Comment from Reviewer 4 pointing out the missing references in row 57-58.

Thank you for pointing this out, we fixed it.

(4) Comment from Reviewer 4 suggesting more explains and show the results of the screening compounds.

As suggested by the reviewer, we added additional explanations in lines 65-73 and showed the screening results in Figure 1—figure supplement 1F-H.

(5) Comment from Reviewer 4 suggesting an in-depth discussion of the findings.

Thank you for the suggestion, we included additional discussion at the end of the article.

(6) Comment from Reviewer 4 suggesting a conclusion should be inculded in the main text.

Thank you for the suggestion, we made a revision.

(7) Comment from Reviewer 4 pointing out the cell viability under different concentrations of 2C.

As mentioned by the reviewer, have supplemented the cell numbers during different doses of 2C treatment (Figure 2F).

(8) Comment from Reviewer 4 pointing out the missing information in the methods.

Thank you for the suggestion, we made additions.

(9) Comment from Reviewer 4 suggesting more explanations in Figure S3A.

As mentioned by the reviewer, we made a revision in original Fig.S3A (now is Figure 2—figure supplement 1).

(10) Comment from Reviewer 4 pointing out the high variability of mCherry cells (%) in Figure 3J.

Thank you. We made a revision.

(11) Comment from Reviewer 4 suggesting more explanations on the DNA-binding motif of ISL1 in the cells treated with A-485 or 2C.

Thank you for the suggestion, we added additional explanations (lines 270-274 on page 9).

(12) Comment from Reviewer 4 pointing out the unclear labeling in Figure S1B and D.

Thank you for the suggestion, made a revision (lines 240-245 in supporting information).

(13) Comment from Reviewer 4 suggesting a relative quantification of the proteins in Figure 1H.

Thank you for the suggestion. We have quantified the relative expression levels of proteins in original Fig. 1H. As shown in Figure 1F.

(14) Comment from Reviewer 4 suggesting to provide detailed information in the methodology part about the compounds.

Thank you for the suggestion, we made a revision.

(15) Comment from Reviewer 4 pointing out the insufficient explanations on figure legends.

Thank you for the suggestion, we made a revision.

(16) Comment from Reviewer 4 suggesting more independent experiments to reduce the high variations in “ns” between NC and 2C at 60h+3d shown in Figure 2E and F.

Thank you for the suggestion, we made a revision in Figure 2F.

(17) Comment from Reviewer 4 suggesting a limitations should be provided in the text.

Thank you for the suggestion, we have made provide a limitation statement in the revised manuscript (lines 300-311 on page 10).